# FBXO22 degrades nuclear PTEN to promote tumorigenesis

Meng-Kai Ge[1,4], Na Zhang[1,4], Li Xia[1,4], Cheng Zhang[1], Shuang-Shu Dong[2], Zhan-Ming Li[1], Yan Ji[2], Min-Hua Zheng[3], Jing Sun[3], Guo-Qiang Chen [1✉] & Shao-Ming Shen [1✉]

Nuclear localization of PTEN is essential for its tumor suppressive role, and loss of nuclear PTEN is more prominent than cytoplasmic PTEN in many kinds of cancers. However, nuclear PTEN-specific regulatory mechanisms were rarely reported. Based on the finding that nuclear PTEN is more unstable than cytoplasmic PTEN, here we identify that F-box only protein 22 (FBXO22) induces ubiquitylation of nuclear but not cytoplasmic PTEN at lysine 221, which is responsible for the degradation of nuclear PTEN. FBXO22 plays a tumor-promoting role by ubiquitylating and degrading nuclear PTEN. In accordance, FBXO22 is overexpressed in various cancer types, and contributes to nuclear PTEN downregulation in colorectal cancer tissues. Cumulatively, our study reports the mechanism to specifically regulate the stability of nuclear PTEN, which would provide the opportunity for developing therapeutic strategies aiming to achieve complete reactivation of PTEN as a tumor suppressor.

[1] Department of Pathophysiology, Key Laboratory of Cell Differentiation and Apoptosis of Chinese Ministry of Education, State Key Laboratory of Oncogenes and Related Genes and Chinese Academy of Medical Sciences Research Unit (NO.2019RU043), Shanghai Cancer Institute, Rui-Jin Hospital, Shanghai Jiao Tong University School of Medicine, 200025 Shanghai, China. [2] Shanghai Institutes for Biological Sciences, Chinese Academy of Sciences, 200025 Shanghai, China. [3] Department of Gastrointestinal Surgery, Ruijin Hospital, Shanghai Jiao Tong University School of Medicine, Shanghai, China. [4]These authors contributed equally: Meng-Kai Ge, Na Zhang, Li Xia. ✉email: chengq@shsmu.edu.cn; smshen@shsmu.edu.cn

Phosphatase and tensin homolog on chromosome 10 (PTEN) is a potent tumor suppressor whose loss of function is frequently observed in both inheritable and sporadic cancers[1]. The lipid phosphatase activity to dephosphorylate the PtdIns(3,4,5)P_3 (PIP_3) to PIP_2 and thus depleting cellular PIP_3 is the most extensively known tumor-suppressive function of PTEN[1–3]. As a protein phosphatase, PTEN has been reported to dephosphorylate itself and several other protein substrates to exert its tumor-suppressive functions[4–8]. Besides, PTEN is also known to act as a scaffold protein to exert phosphatase-independent function, which is as significant as its lipid phosphatase activity to tumor suppression[9–12].

It is believed that cytoplasmic PTEN exerts its lipid phosphatase activity, while nuclear PTEN performs various cellular functions largely through lipid–phosphatase-independent mechanisms[10,12,13]. For instance, nuclear PTEN promotes the association of the anaphase-promoting complex/cyclosome (APC/C) with CDC20-like protein 1 (CDH1) to enhance its E3 ligase activity, resulting in the degradation of oncogenic substrates polo-like kinase 1 (PLK1), Aurora A and Cyclin A2[14]. The expressions of these substrates are later used to evaluate the activity of nuclear PTEN[15,16]. Nuclear PTEN also maintains genome stability through multiple mechanisms such as inducing DNA repair protein RAD51 to positively regulate DNA repair through its interaction with centromere protein C[17,18], and recruiting the deubiquitinase OTU domain-containing ubiquitin aldehyde-binding protein 1 (OTUB1) to mediate deubiquitination and stabilization of replication factor A protein 1 (RPA1)[19]. Also, nuclear PTEN regulates alternative pre-mRNA splicing events through its association with spliceosome[20] or arginine methylation[21].

Nuclear localization is essential for the tumor suppressive role of PTEN, as a PTEN mutant deficient in nuclear import but with intact phosphatase activity causes Cowden syndrome[22], and the absence of nuclear PTEN was associated with more aggressive cancers[22–26]. Intriguingly, it has long been observed that loss of nuclear PTEN, partial or complete, was more significant than cytoplasmic PTEN in cancers[24–26], suggesting the existence of regulatory mechanisms that specifically regulate the abundance of nuclear PTEN rather than cytoplasmic PTEN. However, such mechanisms have rarely been studied until now. We believe the knowledge of such mechanisms would not only help us to learn the relationship between nuclear PTEN and cancer, but also provide new perspective to the therapeutic reactivation of PTEN which has been suggested as a good candidate for the "tumor suppressor reactivation" approach to cancer treatment[27].

Besides genomic alterations of PTEN, which are frequently observed only in certain cancer types, such as prostate cancer, glioblastoma multiforme, and uterine cancer[28,29], post-translational modifications such as phosphorylation, ubiquitylation, acetylation, oxidation, SUMOylation, PARylation, and arginine methylation also contribute to the loss of PTEN in many cancer types[10,21,29–32]. Particularly, ubiquitin-mediated proteasomal degradation of PTEN represents an important post-translational mechanism to maintain an optimal level of PTEN in physiological conditions. The S phase kinase-associated protein 1 (SKP1)–cullin 1 (CUL1)–F-box protein (SCF) ubiquitin ligase complexes mediate the degradation of a large number of diverse processes-involved regulatory proteins[33]. As the substrate-targeting subunits of SCF ubiquitin ligase complexes, F-box proteins specifically recruit substrates to the SCF core. In mammals, there are ~70 F-box proteins, which generate three families, the F-box and WD40 domain (FBXW) family, the F-box and Leu-rich repeat (FBXL) family and F-box only (FBXO) proteins such as FBXO22. To date, some substrates of FBXO22 have been reported, including BTB and CNC homolog 1 (BACH1)[34], Krueppel-like factor 4[35], P53[36], serine/threonine-protein kinase (STK11 or LKB1)[37] and lysine-specific demethylase 4B (KDM4B)[38]. The tumor promoting role of FBXO22 has been presented by some reports in liver cancer[35], lung cancer[37], and breast cancer[39], while an inhibitory role towards cancer metastasis has also been proposed in lung cancer[34] and breast cancer[39]. Herein we report that FBXO22 ubiquitylates and targets nuclear but not cytoplasmic PTEN for proteasome-mediated degradation, through which FBXO22 contributes to nuclear PTEN downregulation in colorectal cancer and exerts tumorigenic function.

## Results

**Nuclear PTEN is sensitive to ubiquitin-mediated degradation.** In our preliminary experiments, we observed that only a small portion of transiently expressed C-terminal green fluorescent protein (GFP)-tagged PTEN (PTEN-GFP) in PTEN knockout human embryonic kidney 293T cells showed nuclear localization, which was dramatically increased in the presence of proteasome inhibitor MG132 (Fig. 1a). Cellular fractionation followed by immunoblotting confirmed the accumulative effect of MG132 on the nuclear portion of PTEN-GFP (Supplementary Fig. 1a). In contrast, MG132 treatment failed to significantly affect the cytoplasmic portion of PTEN-GFP (Supplementary Fig. 1a). Based on this interesting finding, we asked whether proteasome inhibitors also accumulate endogenous nuclear PTEN. As for this, 293T cells and mouse embryonic fibroblasts (MEF) were treated with proteasome inhibitors MG132 and Bortezomib (PS341), along with the lysosome inhibitor NH_4Cl and the endoplasmic reticulum to Golgi transport inhibitor Brefeldin A (BFA), as PTEN had been reported to be degraded by Sirtuin 4 through lysosome pathway[40]. The results showed that both MG132 and Bortezomib abundantly accumulated nuclear PTEN without affecting cytoplasmic PTEN in both cell lines, while NH_4Cl or BFA showed a weak or no effect on nuclear PTEN accumulation (Fig. 1b). Notably, a previous report revealed that mono-ubiquitylation of PTEN on K13 and K289 regulates its nuclear import[22]. Here we showed that MG132 treatment still increased GFP-tagged K13 and K289 mutated PTEN (PTEN^{K13,289E}) in the nucleus to similar degree as wild-type PTEN (PTEN^{WT}, Supplementary Fig. 1b), excluding the possibility that MG132-increased nuclear PTEN was due to its monoubiquitylation. To further validate that MG132 induces stabilization of nuclear PTEN, 293T cells, and MEF were treated with cycloheximide (CHX) for indicated times and subjected to cellular fractionation. The results showed that while total and cytoplasmic PTEN protein hardly decreased by 12 h CHX treatment, consistent with a previous report[32], nuclear PTEN showed a half-life as short as ~3 h in both cell lines (Fig. 1c). To be noted, total PTEN levels in all the above experiments seemed to be unaffected by the change of nuclear PTEN, which might attribute to the relatively small proportion of nuclear PTEN.

To find out whether ubiquitylation is the underlying mechanism for the instability of nuclear PTEN, Flag-tagged PTEN was co-transfected with hemagglutinin (HA)-tagged ubiquitin (Ub) into 293T cells, which were collected, fractionated, and subjected to immunoprecipitation (IP) with anti-Flag antibody. Immunoblotting for HA-Ub-conjugated proteins revealed that nuclear PTEN was more abundantly ubiquitylated than cytoplasmic PTEN, and MG132 treatment significantly accumulated ubiquitylated PTEN in the nucleus, where PTEN was predominantly ubiquitylated by K48 but not K63 ubiquitin chain (Fig. 1d). Taken together, our results indicate that nuclear PTEN is more susceptible than cytoplasmic PTEN to ubiquitin-mediated proteasomal degradation.

**FBXO22 ubiquitylates nuclear rather than cytoplasmic PTEN.** To find ubiquitin ligases specifically targeting nuclear PTEN, we

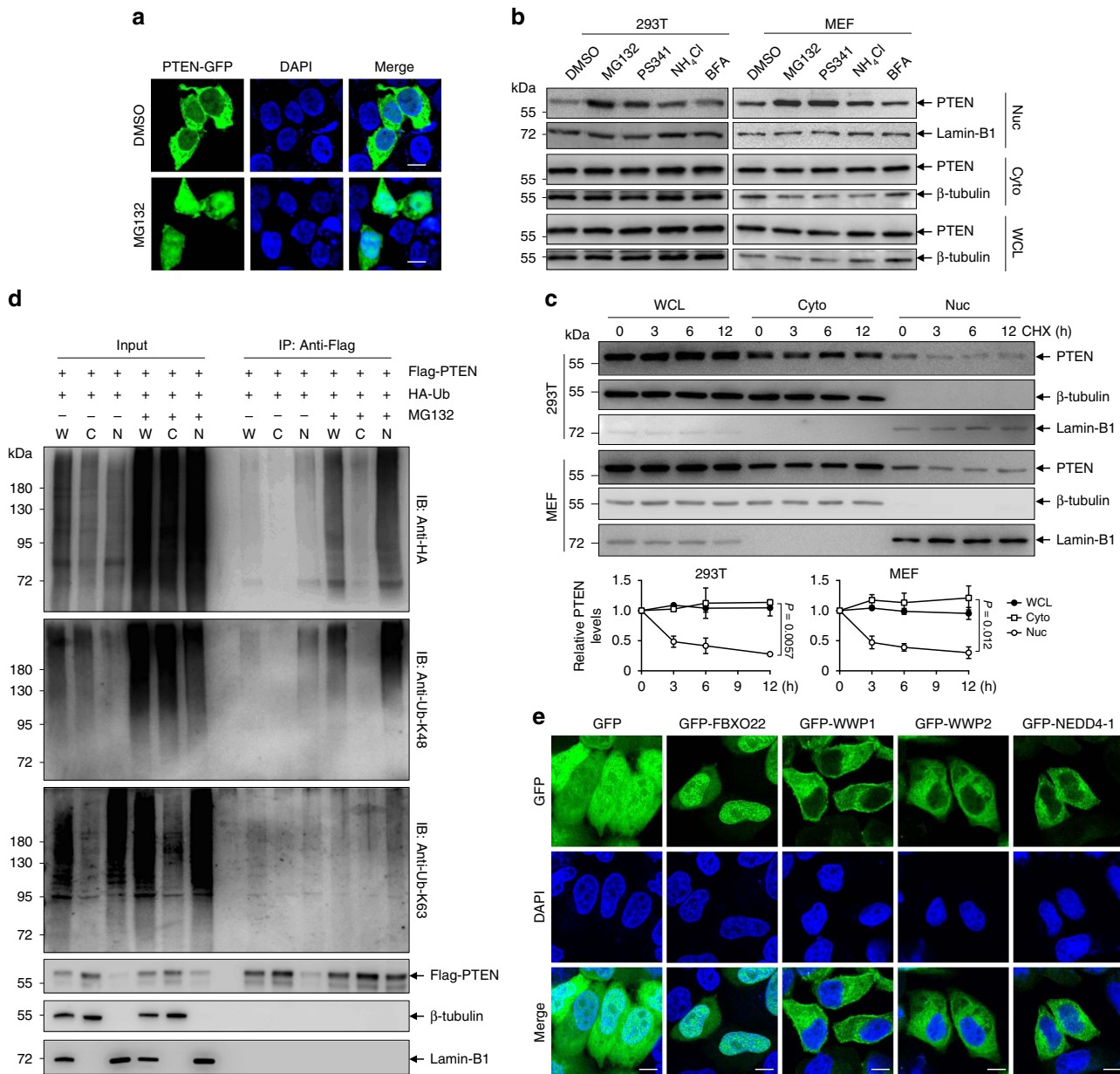

**Fig. 1 Nuclear PTEN is sensitive to ubiquitin-mediated proteasomal degradation. a** A clonally derived 293T PTEN knockout cell line (293T-PTEN$^{KO}$) was transfected with PTEN-GFP for 8 h, followed by treatment with 10 μM MG132 or DMSO for 6 h, stained with DAPI, and imaged with confocal microscopy. Scale bar represents 10 μm. **b** Western blots of indicated proteins in cytoplasmic (Cyto) and nuclear (Nuc) fractions as well as whole-cell lysates (WCL) of 293T cells and MEF treated with DMSO, 10 μM MG132, 10 μM PS341, 10 mM NH$_4$Cl and 10 μM BFA for 8 h. β-tubulin and Lamin-B1, respectively, serve as cytoplasmic and nuclear markers. **c** Western blots of indicated proteins (top) and quantification of relative PTEN levels (bottom) in cytoplasmic and nuclear fractions as well as whole-cell lysates of 293T cells and MEF treated with 50 μg/ml CHX for indicated times. Quantification of protein bands was performed using ImageJ software. **d** 293T cells transfected with the indicated plasmids were treated with 10 μM MG132 or DMSO for 6 h, harvested and submitted to subcellular fractionation, followed by in vivo ubiquitination assay and western blot analysis for indicated proteins. C cytoplasm; N nucleus; W whole-cell lysates. **e** Representative images of GFP-tagged FBXO22, NEDD4-1, WWP2, and WWP1 transfected in Hela cells with re-staining of DAPI. Scale bar represents 10 μm. The experiments shown in **a**, **b**, **d**, and **e** were repeated three times with similar results, and the results of one representative experiment are shown. The data in panel **c** are presented as the mean ± SEM, $n = 3$ independent experiments, statistical significance was determined by two-way ANOVA. Source data are provided as a Source Data file.

firstly tested the possible involvement of NEDD4-1 and WWP2, two known proto-oncogenic ubiquitin ligases for PTEN degradation[16,41]. However, both enzymes showed predominant cytoplasmic localization (Supplementary Fig. 1c), and their knockdown by shRNA increased total and cytoplasmic but not nuclear PTEN in 293T and SW620 cells (Supplementary Fig. 1d), suggesting that these two ligases did not contribute to the degradation of nuclear PTEN. Thus, 293T cells were transiently

transfected with plasmid encoding Flag-tagged PTEN along with empty vector, followed by affinity purification using an anti-Flag antibody, and the bound proteins were analyzed by liquid chromatography with tandem mass spectrometry (LC–MS/MS). We noticed that FBXO22 was the only ubiquitin ligase that presented in the PTEN immunoprecipitate (Supplementary Data 1). More importantly, in line with a previous report[36], GFP-tagged FBXO22 was predominantly nuclear localized in Hela cells, in

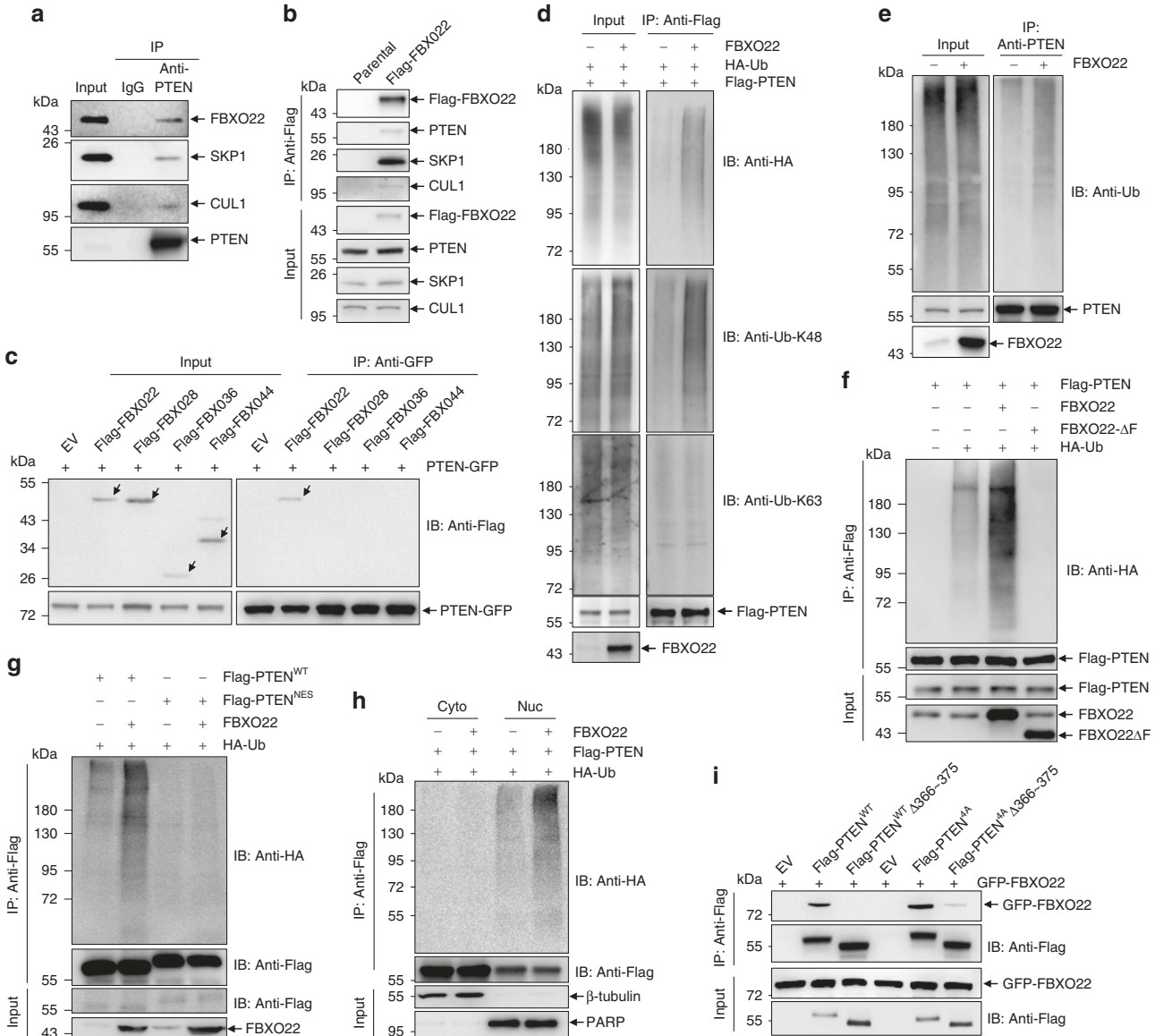

**Fig. 2 FBXO22 ubiquitylates nuclear but not cytoplasmic PTEN. a** Western blots of indicated proteins in the input and immunoprecipitates of IgG or anti-PTEN antibody in 293T cells. **b** 293T cells with a Flag tag knocked into the FBXO22 locus together with parental 293T cells were subjected to immunoprecipitation with an anti-Flag antibody. Western blots of indicated proteins in the input and immunoprecipitates are shown. **c** 293T cells co-transfected with PTEN-GFP and Flag-tagged proteins were subjected to immunoprecipitation with an anti-GFP antibody. Western blots of Flag- and GFP-tagged proteins in the input and immunoprecipitates are shown. **d–g** 293T cells transfected with the indicated plasmids were treated with 10 μM MG132 for 6 h, harvested, and submitted to in vivo ubiquitination assay, followed by Western blot analysis with antibodies as indicated. **h** 293T cells transfected with the indicated plasmids were treated with 10 μM MG132 for 6 h, harvested, and submitted to subcellular fractionation into cytoplasmic (Cyto) and nuclear (Nuc) fractions, followed by in vivo ubiquitination assay and Western blot analysis. **i** 293T cells co-transfected with GFP-FBXO22 and Flag-tagged PTEN derivatives were subjected to immunoprecipitation with an anti-Flag antibody. Western blots of Flag- and GFP-tagged proteins in the input and immunoprecipitates are shown.The experiments shown in **a–i** were repeated three times with similar results, and the results of one representative experiment are shown. Source data are provided as a Source Data file.

contrast to the cytoplasmic localized NEDD4-1, WWP2, and WWP1, a recently discovered nondegradative ubiquitin E3 ligase of PTEN[27] (Fig. 1e).

We continued to investigate the relationship between FBXO22 and PTEN. Our IP assay with an anti-PTEN antibody validated the endogenous interaction of PTEN with FBXO22 and the other two components of the SCF complex, SKP1 and CUL1, in 293T cells (Fig. 2a). Because anti-FBXO22 antibody here used was not suitable for IP, a Flag tag was knocked into the FBXO22 locus of 293T cells, and the resultant endogenous Flag-tagged FBXO22 could immunoprecipitate endogenous PTEN as well as SKP1 and

CUL1 (Fig. 2b). Also, the ectopically expressed C-terminal Flag-tagged CUL1 pulled down endogenous PTEN and FBXO22 in 293T cells (Supplementary Fig. 2a). In addition, FBXO22 but not several other F-box only proteins bound to PTEN (Fig. 2c). Accordingly, co-expression of FBXO22 with PTEN in the presence of HA-Ub led to increased ubiquitylation of PTEN protein by K48 but not K63 ubiquitin chain (Fig. 2d). FBXO22 overexpression also promoted ubiquitylation of endogenous PTEN with endogenous Ub (Fig. 2e). It has been well documented that F-box proteins link to the SCF complex by directly binding SKP1 through the F-box domain[33]. Indeed,

deletion of F-box domain of FBXO22 (FBXO22ΔF) abrogated its interaction with SKP1 but retained the interaction with PTEN (Supplementary Fig. 2b). Consistently, FBXO22ΔF not only failed to promote the ubiquitylation of PTEN, but also seemed to play a dominant-negative role (Fig. 2f). Further, forced localization of PTEN to the cytoplasmic compartment by fusing a nuclear export signal to its C-terminus (PTEN$^{NES}$, Supplementary Fig. 2c) almost completely abolished the effect of FBXO22 (Fig. 2g), suggesting that FBXO22-mediated ubiquitylation of PTEN was dependent on the nuclear localization of PTEN. To further address whether FBXO22 selectively ubiquitylated nuclear PTEN, Flag-tagged PTEN and HA-Ub were co-transfected into 293T cells with or without FBXO22 overexpression. After treatment with MG132 for 6 h, cells were collected, fractionated, and subjected to IP with anti-Flag antibody. Immunoblotting for HA-Ub-conjugated proteins revealed that FBXO22 increased the ubiquitylation of nuclear but not cytoplasmic PTEN (Fig. 2h). Therefore, we identify FBXO22 as a ubiquitin ligase that ubiquitylates nuclear but not cytoplasmic PTEN.

We continued to map the docking region of PTEN for its interaction with FBXO22, which was carried out on the basis of nuclear localized PTEN derivatives. Previously, it was shown that PTEN mutations that affected combinations of the casein kinase 2-phosphorylated residues, including S380/T382/T383/S385, favored PTEN nuclear accumulation[42]. We also showed that S380/T382/T383- phosphorylated PTEN was mainly detected in the cytoplasm of 293T cells and MEF (Supplementary Fig. 2d), while mutation of S380/T382/T383/S385 to alanines (PTEN$^{4A}$) almost completely relocalized PTEN into the nucleus (Supplementary Fig. 2e). In accordance, PTEN$^{4A}$ showed more abundant interaction with FBXO22, compared to PTEN$^{WT}$ (Supplementary Fig. 2f). Meanwhile, C-tail-deleted PTEN (PTEN$^{1\sim351}$) also manifested predominant nuclear localization (Supplementary Fig. 2e, g), while it only showed a weak binding with FBXO22 (Supplementary Fig. 2h), indicating the C-tail of PTEN was required for the efficient interaction with FBXO22. Further analysis revealed that the predominantly nuclear localized fragment PTEN$^{1\sim365}$ did not, but PTEN$^{1\sim375}$ could interact with FBXO22 to the similar degree as PTEN$^{4A}$ (Supplementary Fig. 2e, h). Based on these findings, we confirmed that aa 366~375 of PTEN is critical for its interaction with FBXO22 (Fig. 2i).

**FBXO22 destabilizes nuclear but not cytoplasmic PTEN.** The above observations promoted us to investigate whether FBXO22 regulates the abundance of nuclear PTEN. For this purpose, we generated stable pool cells with FBXO22 depletion, either by lentiviruses encoding CAS9 and a guide RNA targeting *FBXO22* gene locus (gFBXO22) with gNS as a non-specific control (Fig. 3a), or by retroviruses encoding FBXO22-specific shRNA (shFBXO22) with shNC as a negative control (Fig. 3b), in two colorectal cancer cell lines, SW620 and SW480. The tumor suppressive function of nuclear PTEN in colon cancer has been previously explored[19,22], and genetic alteration of *PTEN* gene is relatively rare in colon cancer[28], suggesting that regulation of PTEN more likely relies on mechanisms beyond genetic alterations in this kind of cancer. Thus, we chose colon cancer as a model for our further study. Depletion of FBXO22 by either CRISPR-CAS9 (Fig. 3a) or shRNA (Fig. 3b) in both cell lines increased the levels of nuclear but not cytoplasmic PTEN, as well as a known FBXO22 substrate BACH1[34]. Here we further showed that unlike FBXO22, knockdown of NEDD4-1 and WWP2 by their specific shRNAs only increased cytoplasmic PTEN without influencing nuclear PTEN (Supplementary Fig. 3a). Of note, increases of total PTEN were also observed by FBXO22 depletion (Fig. 3a, b), probably due to the persistent accumulation of

nuclear PTEN during propagation of stable FBXO22-depleted cells. To achieve controllable FBXO22 overexpression, on the other hand, SW620 cell line with inducible FBXO22 expression by a doxycycline (DOX)-inducible expression system (SW620-FBXO22$^{ind}$) was generated. After FBXO22 protein was gradually induced by either a time course or a gradient of DOX treatment, we observed a progressive decrease of BACH1 and nuclear PTEN (Fig. 3c), the latter being blocked by treatment with MG132 (Supplementary Fig. 3b). Similar results were obtained in LS174T cells (Supplementary Fig. 3c). All these results indicate that FBXO22-mediated ubiquitylation leads to degradation of nuclear PTEN.

We further demonstrated that a highly probable sequence, K$^{121}$RARKR$^{126}$, is the nuclear localization signal (NLS) of FBXO22, because its mutation (FBXO22$^{mNLS}$) almost completely excluded FBXO22 from the nucleus (Fig. 3d). Considering that FBXO22 was also localized in cytoplasm to a degree (Figs. 1e and 3d), we asked whether cytoplasm-localized FBXO22 was capable of degrading cytoplasmic PTEN. For this purpose, FBXO22$^{mNLS}$ along with FBXO22$^{WT}$ were inducibly expressed in SW620 cells, and the results showed that overexpression of FBXO22$^{WT}$ downregulated nuclear but not cytoplasmic PTEN, while FBXO22$^{mNLS}$ affected neither nuclear nor cytoplasmic PTEN (Fig. 3e). Further analysis revealed that the inability of FBXO22$^{mNLS}$ to degrade cytoplasmic PTEN might attribute to their inability to interact with each other, as FBXO22$^{mNLS}$ failed to interact with PTEN$^{WT}$ (Fig. 3f).

To evaluate the extent to which nuclear PTEN instability was dependent on FBXO22, two clonally derived 293T FBXO22 knockout cell lines generated by CRISPR-CAS9 (293T-FBXO22$^{KO-1}$ and 293T-FBXO22$^{KO-2}$), along with a clonally derived control cell line generated by using a non-specific (NS) guide RNA (293T-NS$^{clone}$) were treated with CHX for indicated time points, and the stabilities of total, cytoplasmic and nuclear PTEN were compared. In NS$^{clone}$ cells, total and cytoplasmic PTEN hardly decreased by 12 h CHX treatment, while nuclear PTEN showed a much shorter half life (Fig. 3g), similar with the observations made in parental 293T cells (Fig. 1c). However, the decrease of nuclear PTEN under CHX treatment was almost completely blocked in FBXO22$^{KO-1}$ and FBXO22$^{KO-2}$ cells (Fig. 3g). Similar results were also obtained in two colorectal cancer cell lines SW620 and LS174T (Supplementary Fig. 3d). All these results support that FBXO22 is a major factor regulating the stability of nuclear PTEN.

**FBXO22 inhibits nuclear PTEN action to promote tumorigenesis.** In accordance with the notion that FBXO22 degraded nuclear PTEN, FBXO22 depletion by shRNA (Fig. 4a) or CRISPR-CAS9 (Fig. 4b) showed no impact on AKT activation, but significantly reduced the expressions of APC/C–CDH1 substrates, PLK1, Aurora A and Cyclin A2, which could be reversed by ectopic expression of a shRNA-resistant FBXO22 (Supplementary Fig. 4a). As a result, FBXO22 knockdown induced G$_1$ arrest in SW620 cells (Supplementary Fig. 4b). On the contrary, knockdown of both NEDD4-1 and WWP2 inhibited AKT activation with no inhibitory effect on PLK1, Aurora A or Cyclin A2 (Supplementary Fig. 4c). It is worthwhile to point out that knockdown of NEDD4-1 appeared to increase PLK1 (Supplementary Fig. 4c), but the underlying mechanism remains to be illustrated, although a previous report showed that PLK1 also directly phosphorylates Nedd4-1[43].

On the other hand, inducible expression of FBXO22 increased the expressions of PLK1, Aurora A and Cyclin A2 in SW620 cells (Fig. 4c), which could be blocked by ectopic expression of CDH1 (Supplementary Fig. 4d), suggesting that FBXO22 regulates these

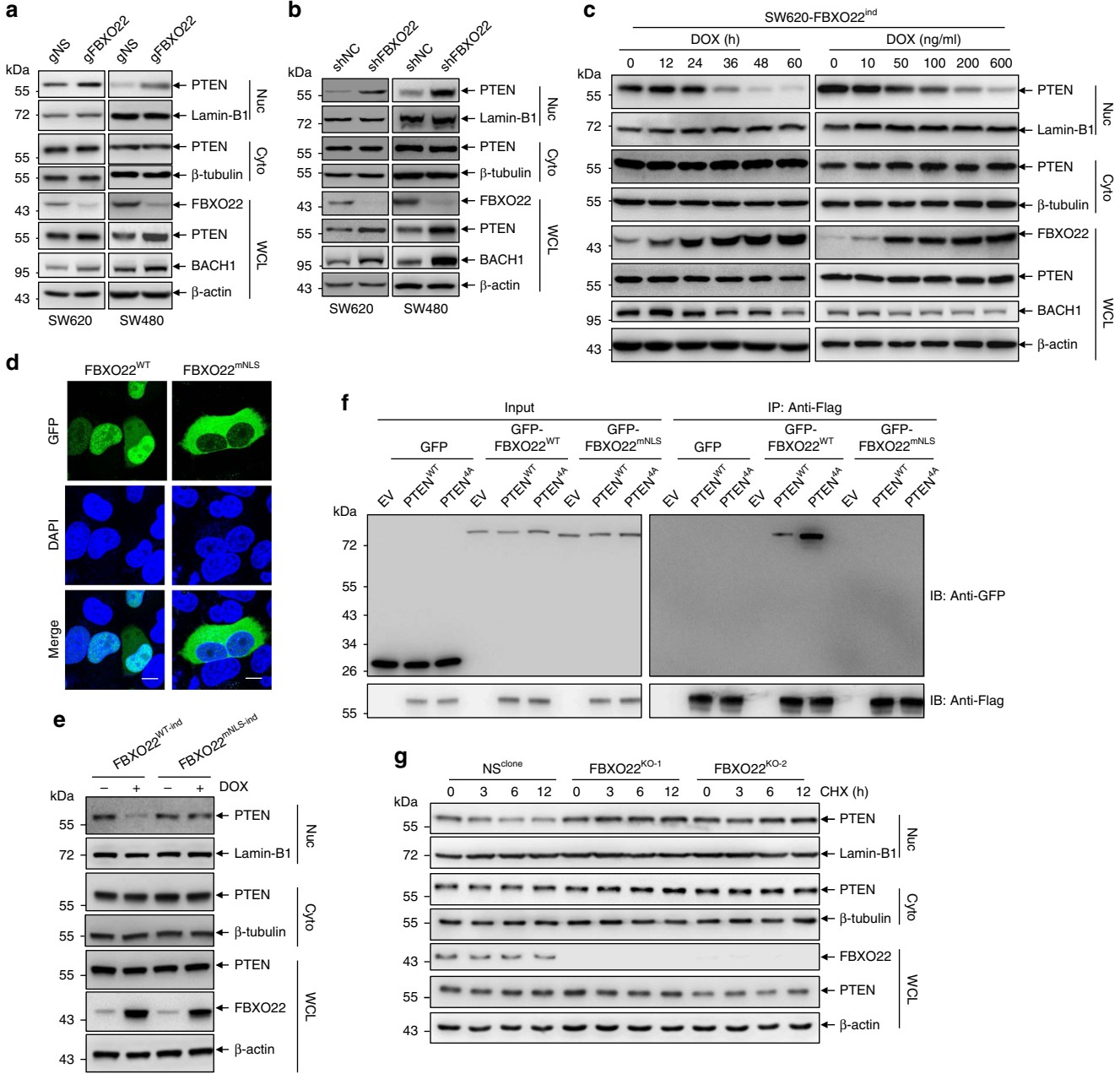

**Fig. 3 FBXO22 destabilizes nuclear but not cytoplasmic PTEN. a**, **b** Western blots of indicated proteins in the nuclear (Nuc) and cytoplasmic (Cyto) fractions as well as whole-cell lysates (WCL) of SW620 or SW480 cells transduced by lentiviruses encoding CAS9 and gFBXO22, along with gNS as a non-specific control (**a**), and retroviruses encoding a shRNA targeting FBXO22 (shFBXO22), along with shNC as a negative control (**b**). **c** A doxycycline (DOX)-inducible expression system encoding FBXO22 was introduced by lentiviruses into SW620 cells (SW620-FBXO22[ind]). Western blots of indicated proteins in the nuclear and cytoplasmic fractions as well as whole-cell lysates of SW620-FBXO22[ind] cells by DOX administration for indicated times (left) or concentrations (right). **d** Representative images of GFP-tagged FBXO22[WT] and FBXO22[mNLS] in Hela cells with re-staining of DAPI. Scale bar represents 10 μm. **e** A DOX-inducible expression system encoding FBXO22[WT] or FBXO22[mNLS] was introduced by lentiviruses into SW620 cells (SW620-FBXO22[WT-ind] or SW620-FBXO22[mNLS-ind]). After DOX administration, cells were fractionated and subjected to Western blots of indicated proteins. **f** 293T cells co-transfected with GFP, GFP-FBXO22[WT] or GFP-FBXO22[mNLS] and Flag-tagged PTEN[WT] or PTEN[4A] were subjected to immunoprecipitation with an anti-Flag antibody. Western blots of Flag- and GFP-tagged proteins in the immunoprecipitates are shown. **g** Western blots of indicated proteins in the nuclear and cytoplasmic fractions as well as whole-cell lysates of two clonally derived 293T cell lines depleted of FBXO22 (FBXO22[KO-1] and FBXO22[KO-2]) and a negative control (NS[clone]) respectively treated with 50 μg/ml CHX for hours as indicated. The experiments shown in **a**–**g** were repeated three times with similar results, and the results of one representative experiment are shown. Source data are provided as a Source Data file.

proteins through APC/C–CDH1. Inducible expression of FBXO22 also increased the expressions of PLK1 and Aurora A in LS174T cells, but the effect on Cyclin A2 was mild, probably due to the high basal level of Cyclin A2 in this cell line (Supplementary Fig. 4e). Previously, we reported that nuclear

PTEN regulates the efficiency of alternative splicing of cancer-related genes, including GOLGA2, whose splicing promotes Golgi extension and secretion[20]. We thus tested whether FBXO22 also plays a role in alternative splicing. For this purpose, the pTWT minigene which contains α-tropomysin (α-TM) exons 1, 3, and

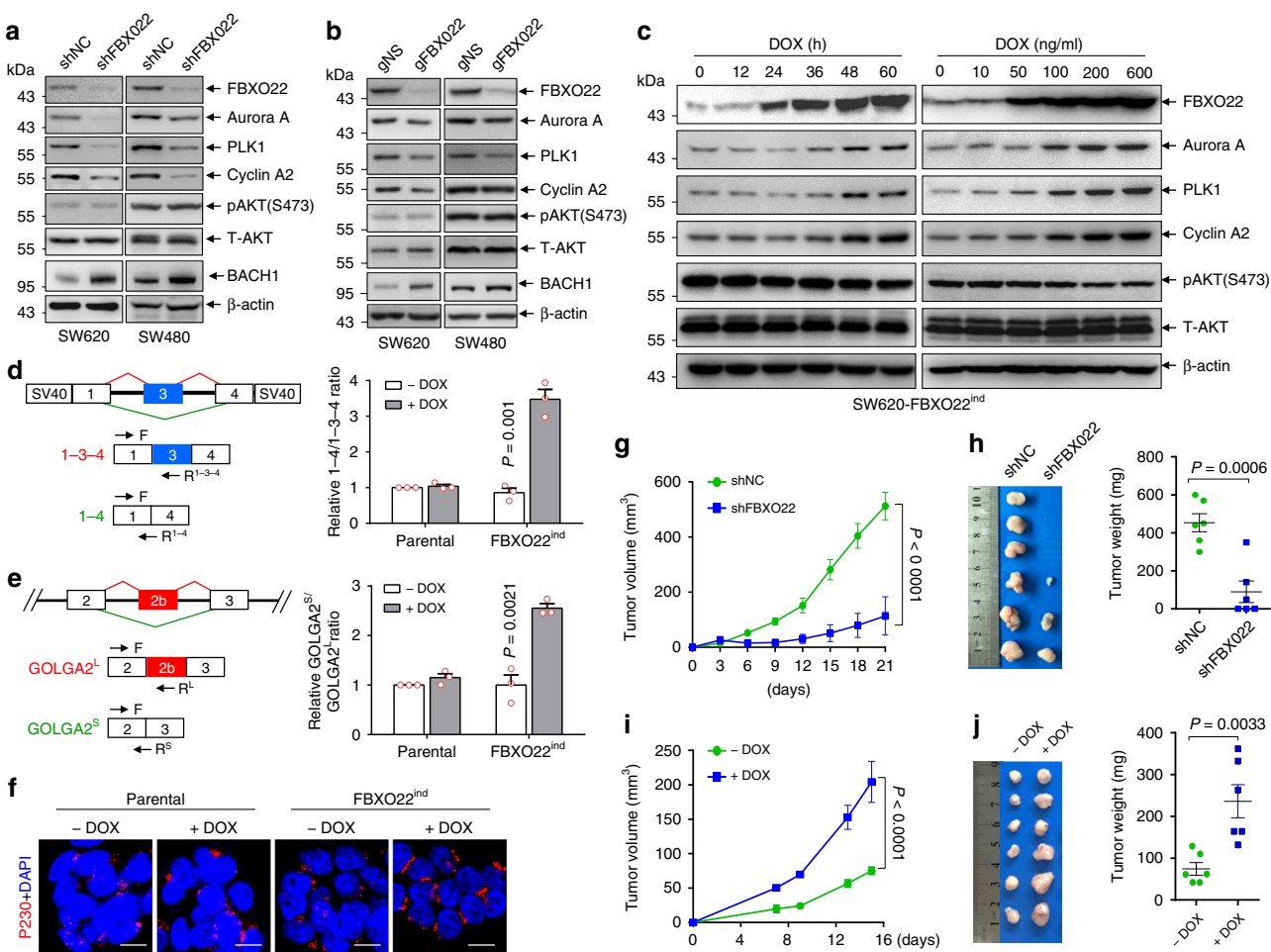

**Fig. 4 FBXO22 suppresses nuclear PTEN signaling and promotes tumorigenesis. a**, **b** Western blots of indicated proteins in SW620 and SW480 cells transduced by retroviruses encoding shFBXO22 or shNC (**a**) and lentiviruses encoding CAS9 and gFBXO22 or gNS (**b**). **c** Western blots of indicated proteins in SW620-FBXO22ind cells by DOX administration for indicated times (left) or concentrations (right). **d**, **e** Left, schematic representation of pTWT reporter gene (**d**) and a part of GOLGA2 gene (**e**) with exons shown as boxes. Splicing patterns are shown as diagonal red (included) or green (excluded) lines, with the two corresponding splicing products shown below. Arrows indicate the locations of primers used in qRT-PCR. Right, qRT-PCR analysis of indicated splicing ratios in parental SW620 or SW620-FBXO22ind cells with or without DOX administration. **f** Representative images of immunofluorescent staining of p230 together with re-staining of DAPI in SW620-FBXO22ind cells with or without FBXO22 induction by DOX administration. Scale bar represents 10 μm. **g**, **h** SW620 cells with or without FBXO22 knockdown by shRNA were subcutaneously injected into nude mice ($3 \times 10^6$ cells per mouse). Tumor volumes were measured at different days (**g**). On day 21 after subcutaneous injection, tumors were harvested and weighted (**h**). **i**, **j** SW620-FBXO22ind cells were subcutaneously injected into nude mice ($1.5 \times 10^6$ cells per mouse). Mice were administrated with or without 600 mg/kg DOX in feed immediately after subcutaneous injection. Tumor volumes were measured at different days (**i**). On day 15 after subcutaneous injection, tumors were harvested and weighted (**j**). The experiments shown in **a–c**, **f** were repeated three times with similar results, and the results of one representative experiment are shown. The data in **d**, **e** are presented as the mean ± SEM, $n = 3$ independent experiments; statistical significance was determined by two-tailed unpaired $t$-test and the $P$ value is shown. The data in **g–j** are presented as the mean ± SEM, $n = 6$ biologically independent samples; two-way ANOVA for **g**, **i** and two-tailed unpaired $t$-test for **h**, **j**. Source data are provided as a Source Data file.

4 surrounded by flanking regulatory sequences and generates two mRNA isoforms, 1-3-4 and 1-4 by alternative splicing (Fig. 4d, left)[44], or the double-reporter minigene pTN24 in which β-galactosidase (β-gal) is constitutively express while luciferase is expressed only if appropriate splicing removes an upstream intron that contains translational stop codons (Supplementary Fig. 4f, top)[45], were transiently transfected into SW620-FBXO22ind cells. The results showed that the induced expression of FBXO22 by DOX treatment caused a decreased splicing ratio of both minigenes, as detected respectively by the increased ratio of exon 3 excluded/included isoform (Fig. 4d) and the decreased luciferase/β-gal ratio (Supplementary Fig. 4f), indicating that FBXO22 regulated the splicing of these minigenes. Moreover, overexpression of FBXO22 induced the skipping of GOLGA2

exon 2b (Fig. 4e), leading to the extension of Golgi Apparatus as assessed by immunofluorescent staining of p230 (Fig. 4f), mirroring the suppressive role of PTEN on GOLGA2 exon 2b skipping and Golgi extension[20]. All these results indicate that FBXO22 regulates the downstream signaling of nuclear PTEN.

To evaluate the role of FBXO22 on tumorigenesis, we showed that knockdown of FBXO22 in SW620 cells by shRNA dramatically suppressed subcutaneous tumor growth (Fig. 4g, h). On the other hand, after subcutaneous injection of SW620-FBXO22ind cells, mice were administrated with or without DOX in feed according to a previous report[46]. DOX administration induced tumoral FBXO22 overexpression (Supplementary Fig. 4g) and promoted tumor growth (Fig. 4i, j). Thus, FBXO22 plays a tumor-promoting role.

**FBXO22 ubiquitylates PTEN on lysine 221**. We then asked whether the FBXO22-mediated inhibition of nuclear PTEN and its downstream signaling are connected to the tumor-promoting role of FBXO22. Toward this end, we firstly sought to localize the FBXO22 ubiquitylated site(s) in PTEN by using a stepwise mutation strategy. As shown in Supplementary Fig. 5a, we divided 34 lysines in PTEN amino acid sequence into four sections, respectively named section A–D. By respectively mutating all lysines in each section to arginines, which generated four PTEN mutants (mut A–D), we found that mutation of the 9 lysines in section B (mut B) completely abolished PTEN ubiquitylation by FBXO22, while other mutants showed no such effect (Supplementary Fig. 5b). Later, the 9 lysines in section B were further divided into three sub-sections (section B1-3) (Supplementary Fig. 5a), and three mutants were generated by respectively mutating all lysines to arginines in each sub-section (mut B1-3). The results showed that three lysines in sub-section B3 were responsible for PTEN ubiquitylation by FBXO22 (Supplementary Fig. 5c). With mutating each of the lysines in sub-section B3 to arginine, we found that mutation of lysine 221 (K221) almost completely abolished K48 ubiquitin chain linked PTEN ubiquitylation by FBXO22 (Fig. 5a, b), without affecting the interaction between PTEN and FBXO22 (Supplementary Fig. 5d) or the localization of PTEN (Supplementary Fig. 5e), suggesting that K221 was the ubiquitylation site of PTEN by FBXO22. In accordance, PTEN K221 mutant (PTEN$^{K221R}$) stably expressed in SW620-PTEN$^{KO}$ cells was more stable than wild-type PTEN (PTEN$^{WT}$) in the nucleus when treated with CHX (Fig. 5c). To validate the role of PTEN K221, PTEN$^{WT}$ and PTEN$^{K221R}$ were respectively re-expressed in PTEN-deficient SW620-PTEN$^{KO}$ cells with inducible expression of FBXO22 (SW620-PTEN$^{KO}$; FBXO22$^{ind}$). When FBXO22 over-expression was induced by DOX (Fig. 5d), we observed down-regulation of nuclear but not cytoplasmic PTEN$^{WT}$, with concomitant increases of PLK1, Aurora A and Cyclin A2, phenocopying the effect of FBXO22 on endogenous PTEN. However, these effects of FBXO22 over-expression were completely resisted by PTEN$^{K221R}$ (Fig. 5d). The same phenomenon was also observed in LS174T cells (Fig. 5e). Thus, our results clearly show that FBXO22 ubiquitylates K221 of PTEN to regulate its stability and downstream signaling.

**FBXO22 promotes tumorigenesis through degrading nuclear PTEN**. We continued to verify the connection between FBXO22-mediated nuclear PTEN degradation and the tumor-promoting role of FBXO22, by utilizing the PTEN$^{K221R}$ mutant. Cell lines respectively expressing empty vector (EV), PTEN$^{WT}$ or PTEN$^{K221R}$, were generated on the basis of SW620-PTEN$^{KO}$; FBXO22$^{ind}$ cells. SW620-NS$^{clone}$;FBXO22$^{ind}$ cells expressing EV were used as control. All these cell lines were subcutaneously injected into nude mice, and mice bearing each cell line were divided into 2 groups with or without DOX administration. In the groups without DOX administration, PTEN deficiency dramatically enhanced tumor growth (group 3 vs group 1), while re-introduction of both PTEN$^{WT}$ and PTEN$^{K221R}$ suppressed tumor growth to similar levels (group 5 or 7 vs group 3) (Fig. 6a, b). As expected, DOX administration almost completely alleviated the growth suppression by PTEN$^{WT}$ (group 6 vs group 5), but showed no such effect on tumors expressing PTEN$^{K221R}$ (group 8 vs 7) (Fig. 6a, b). Successful induction of tumoral FBXO22 overexpression by DOX administration was confirmed (Fig. 6c). Depletion of PTEN increased the levels of PLK1, Aurora A, Cyclin A2, as well as AKT phosphorylation (group 3 vs group 1), which were inhibited by both re-introduction of PTEN$^{WT}$ (group 5 vs group 3) and PTEN$^{K221R}$ (group 7 vs group 3) (Fig. 6c). In PTEN$^{WT}$-re-introduced tumors, induction of FBXO22 increased

the levels of PLK1, Aurora A, and Cyclin A2, but failed to do so in PTEN$^{K221R}$-re-introduced tumors (Fig. 6c). However, induction of FBXO22 failed to affect AKT phosphorylation in both PTEN$^{WT}$- and PTEN$^{K221R}$-re-introduced tumors (Fig. 6c). The similar results were obtained in LS174T cells-derived tumors (Fig. 6d–f). All these results support that FBXO22 promotes tumor growth through K221 ubiquitylation-mediated degradation of nuclear PTEN.

**FBXO22 is negatively correlated with nuclear PTEN in cancers**. Specific loss of nuclear PTEN in cancers is frequently observed by previous reports and contributes to cancer development[24–26]. Thus, we asked whether FBXO22 was one of the reasons that caused nuclear PTEN loss in cancers. For this purpose, we firstly examined the expressional pattern of FBXO22 in all cancer types of The Cancer Genome Atlas (TCGA) database. Intriguingly, in most cancer types, FBXO22 expression was upregulated in cancer tissues compared to normal tissues (Fig. 7a). By comparing samples with 35% highest and lowest FBXO22 expression, we also observed significant inverse correlations between FBXO22 expression and overall patient survival in TCGA datasets of several cancer types (Supplementary Fig. 6a–c). Such a correlation for colon cancer was not observed in TCGA dataset (Supplementary Fig. 6d), but could be observed in Gene Expression Omnibus (GEO) dataset GSE17536 (Fig. 7b). In addition, upregulation of FBXO22 mRNA was confirmed by quantitative real-time PCR (qRT-PCR) in cancer tissues of a cohort of 37 colorectal cancer patients compared to adjacent normal tissues (Fig. 7c, Supplementary Data 2). To investigate the influence of upregulated cancerous FBXO22 on nuclear PTEN in tumor tissues, we also performed immunohistochemistry (IHC) staining with the tissue microarray of human colorectal cancer tissues and paired normal tissues ($n = 72$, Supplementary Data 3). As expected, FBXO22 showed predominant nuclear localization in both cancer and normal tissues (Fig. 7d). In accordance with the above observations, 45 out of 72 samples showed increased FBXO22 in cancer tissues compared to paired normal tissues, while only 4 samples showed decreased FBXO22 (Fig. 7e). Meanwhile, PTEN was localized in both cytoplasm and nucleus in normal tissues, but these two compartments of PTEN underwent distinct cancer-related changes. While 41 out of 72 samples showed decreased nuclear PTEN in cancer tissues compared to paired normal tissues, only 2 samples showed increased nuclear PTEN (Fig. 7e). However, cytoplasmic PTEN showed no obvious alteration trend (Fig. 7e), indicating the specific downregulation of nuclear PTEN in colorectal cancer tissues. Moreover, there was a negative correlation between increased FBXO22 and decreased nuclear PTEN, as 71.11% of the samples with increased FBXO22 expressed decreased nuclear PTEN, while only 33.33% with decreased or equal FBXO22 did (Fig. 7f). Collectively, our findings indicate that overexpression of FBXO22 contributes to downregulation of nuclear PTEN in colon cancer tissues.

## Discussion

During the last decade, the tumor suppressive role of nuclear PTEN has been supported by the discovery of a variety of functions exerted by the nuclear PTEN, mostly independent of its lipid phosphatase activity[10,14,19,20,22]. Despite its importance, the property of nuclear PTEN remains largely unknown. One previous report using signal peptides to respectively target exogenous PTEN to the nucleus and cytoplasm showed that nuclear PTEN was more stable than cytoplasmic PTEN[22]. However, here we showed that endogenous nuclear PTEN was more unstable and sensitive to the ubiquitin-mediated proteasome degradation. The difference might be explained by the fact that exogenously

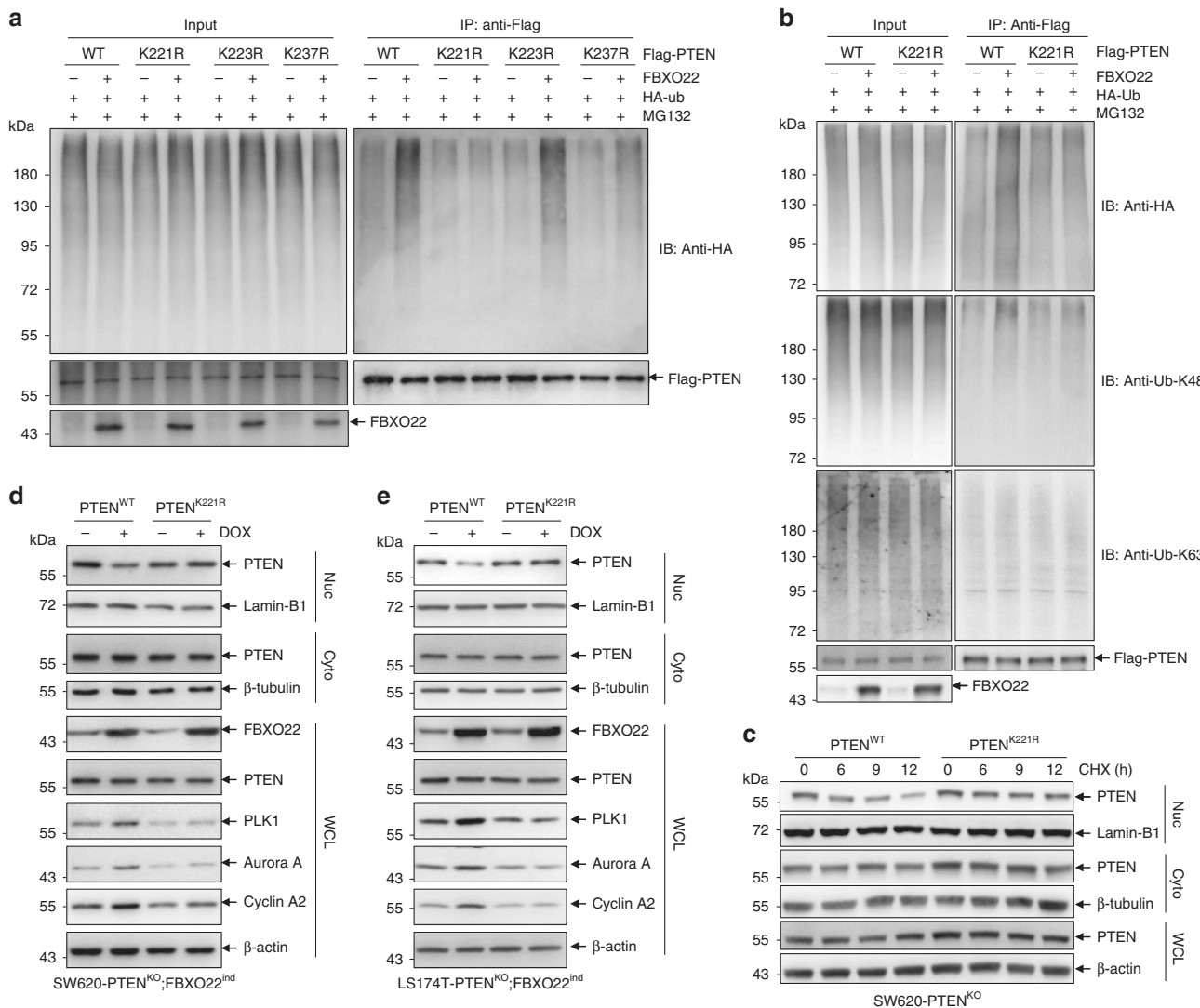

**Fig. 5 FBXO22 ubiquitylates PTEN on lysine 221. a, b** 293T cells transfected with the indicated plasmids were treated with 10 μM MG132 for 6 h, harvested, and submitted to in vivo ubiquitination assay, followed by Western blot analysis with antibodies as indicated. **c** Western blots of indicated proteins in cytoplasmic (Cyto) and nuclear (Nuc) fractions as well as whole-cell lysates (WCL) of PTEN^WT or PTEN^K221R-infected SW620-PTEN^KO cells treated with 50 μg/ml CHX for indicated times. **d, e** A doxycycline (DOX)-inducible expression system encoding FBXO22 was introduced by lentiviruses into the clonally derived PTEN-deficient SW620-PTEN^KO cells (SW620-PTEN^KO;FBXO22^ind) (**d**) and LS174T-PTEN^KO cells (LS174T-PTEN^KO;FBXO22^ind) (**e**), followed by infected by retroviruses encoding wild-type (PTEN^WT) or mutant PTEN (PTEN^K221R). The resultant cell lines were administrated with or without DOX for 12 h, harvested, fractionated, and subjected to western blot analysis of indicated proteins in the nuclear and cytoplasmic fractions as well as whole-cell lysates. The experiments shown in **a–e** were repeated three times with similar results, and the results of one representative experiment are shown. Source data are provided as a Source Data file.

expressed nuclear PTEN might be excessive, especially compared to the small amount of endogenous nuclear PTEN. Also, it is possible that the signal peptide might fail to direct exogenous PTEN to the same location inside the nucleus as endogenous nuclear PTEN, which prevented its association with regulatory proteins. Nevertheless, our results suggested the differential regulation of nuclear and cytoplasmic PTEN stability and the existence of mechanisms that specifically regulated the stability of nuclear PTEN. Actually, there has long been evidence suggesting the existence of such mechanisms. For example, normal thyroid follicular cells are uniformly characterized by a strong staining of nuclear PTEN, whereas the intensity of nuclear PTEN staining progressively diminishes from normal to follicular adenoma to carcinoma and precedes the reduction in cytoplasmic staining which is a characteristic of more advanced thyroid tumors[24]. The nuclear PTEN is also partially or totally lost in almost 80% of

colorectal colon cancer samples, while <30% samples present loss of cytoplasmic PTEN[25]. Among 92 samples from individuals with incident primary cutaneous melanoma, 30 had no or decreased cytoplasmic PTEN protein expression and the remaining 62 had normal cytoplasmic PTEN expression, while in contrast, 84 tumors had no or decreased nuclear expression and 8 had normal nuclear PTEN expression[26]. However, till now, mechanisms that selectively regulate the abundance of nuclear PTEN have rarely been reported. Knockdown of NEDD4-1 and WWP2, two ubiquitin E3 ligases, which had been reported to target PTEN for ubiquitin-mediated proteasome degradation, only affected the cytoplasmic portion of PTEN. Therefore, we set out to look for a new E3 ligase, and identified FBXO22, which selectively ubiquitylated PTEN in the nucleus to accelerate its degradation by proteasome. As a result, unlike NEDD4-1 or WWP2, FBXO22 did not affect AKT activation, but negatively regulated nuclear

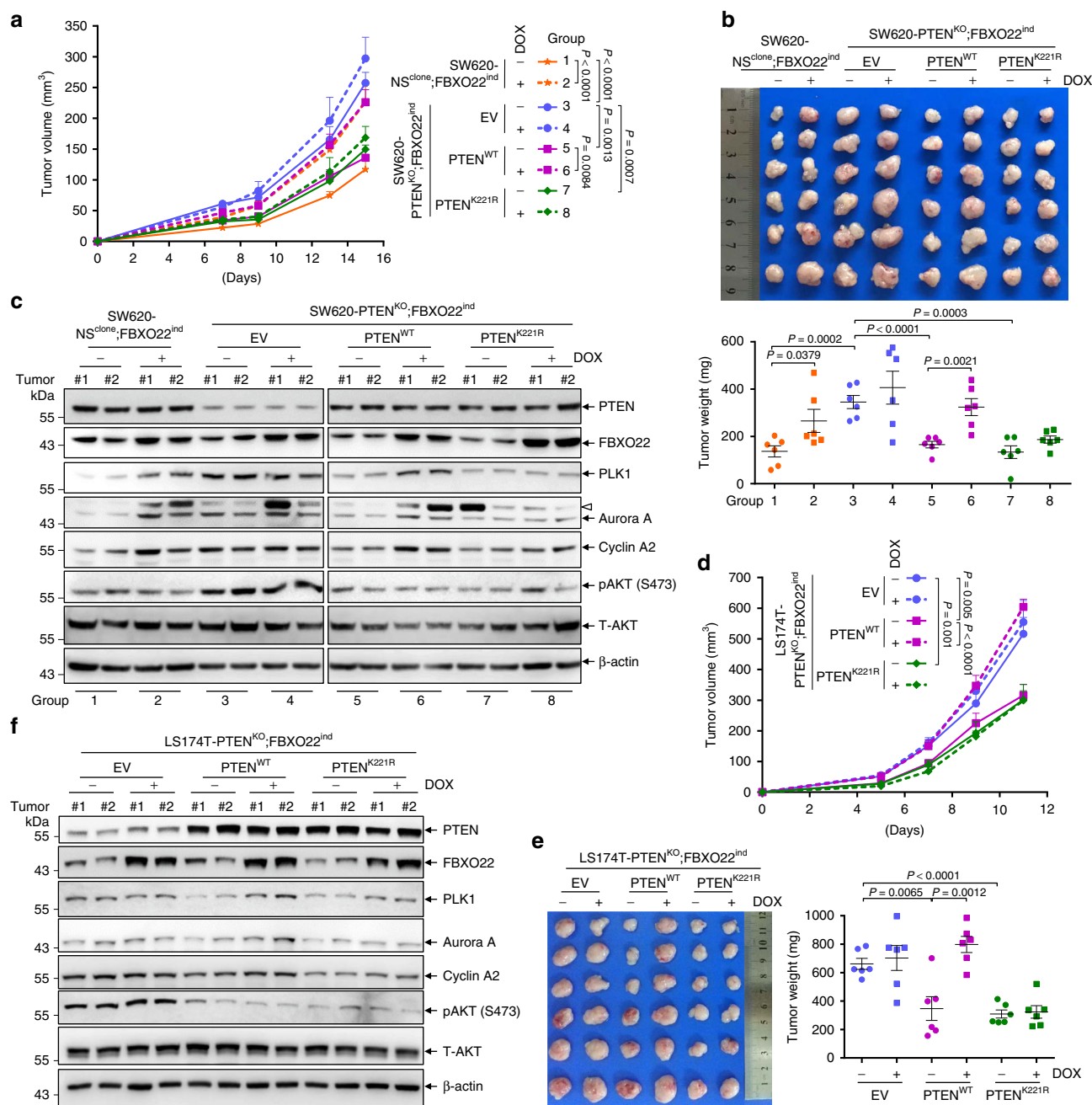

**Fig. 6 FBXO22 exerts tumor-promoting role through degradation of nuclear PTEN. a–c** SW620-NS^clone;FBXO22^ind infected by retroviruses encoding empty vector (EV) and SW620-PTEN^KO;FBXO22^ind cells infected by retroviruses encoding EV, PTEN^WT and PTEN^K221R were subcutaneously injected into nude mice ($1.5 \times 10^6$ cells per mouse). Once inoculated, mice were administrated with or without 600 mg/kg DOX in feed. Tumor volumes were measured at different days (**a**). On day 15 after subcutaneous injection, tumors were harvested, weighted (**b**), and two tumors from each group were randomly selected for Western blot analysis (**c**). The empty triangle points to an unknown band. **d–f** LS174T-PTEN^KO;FBXO22^ind cells were infected by retroviruses encoding EV, PTEN^WT and PTEN^K221R followed by subcutaneous injection into NOD-SCID mice ($1.5 \times 10^6$ cells per mouse). Right after inoculation, mice were administrated with or without 600 mg/kg DOX in feed. Tumor volumes were measured at different days (**d**). On day 11 after subcutaneous injection, tumors were harvested and weighted (**e**), and two tumors from each group were randomly selected for Western blot analysis (**f**). All experiments were repeated three times with similar results, and the results of one representative experiment are shown. The data in panels **a**, **b**, **d**, **e** are presented as the mean ± SEM, $n = 6$ biologically independent samples, and in panels **c**, **f**, tumors ($n = 2$) were randomly selected from each group; two-way ANOVA for **a**, **d** and two-tailed unpaired t-test for **b**, **e**. Source data are provided as a Source Data file.

PTEN functions, such as the activity of APC/C-CDH1 complex and alternative splicing, to play a tumor-promoting role.

By using stepwise mutation, we mapped the ubiquitylated site of PTEN by FBXO22 to be K221. Mutation of K221 did not visibly change the subcellular localization of PTEN. However, it not only resisted the ubiquitylation and degradation of PTEN by

FBXO22, but also almost completely suppressed the tumor-promoting role of FBXO22. In consistence with its tumor-promoting role through downregulating nuclear PTEN, FBXO22 was overexpressed in most types of cancers. Moreover, we confirmed that the absence of nuclear PTEN was more prominent than cytoplasmic PTEN in the cancer tissues of colorectal cancer

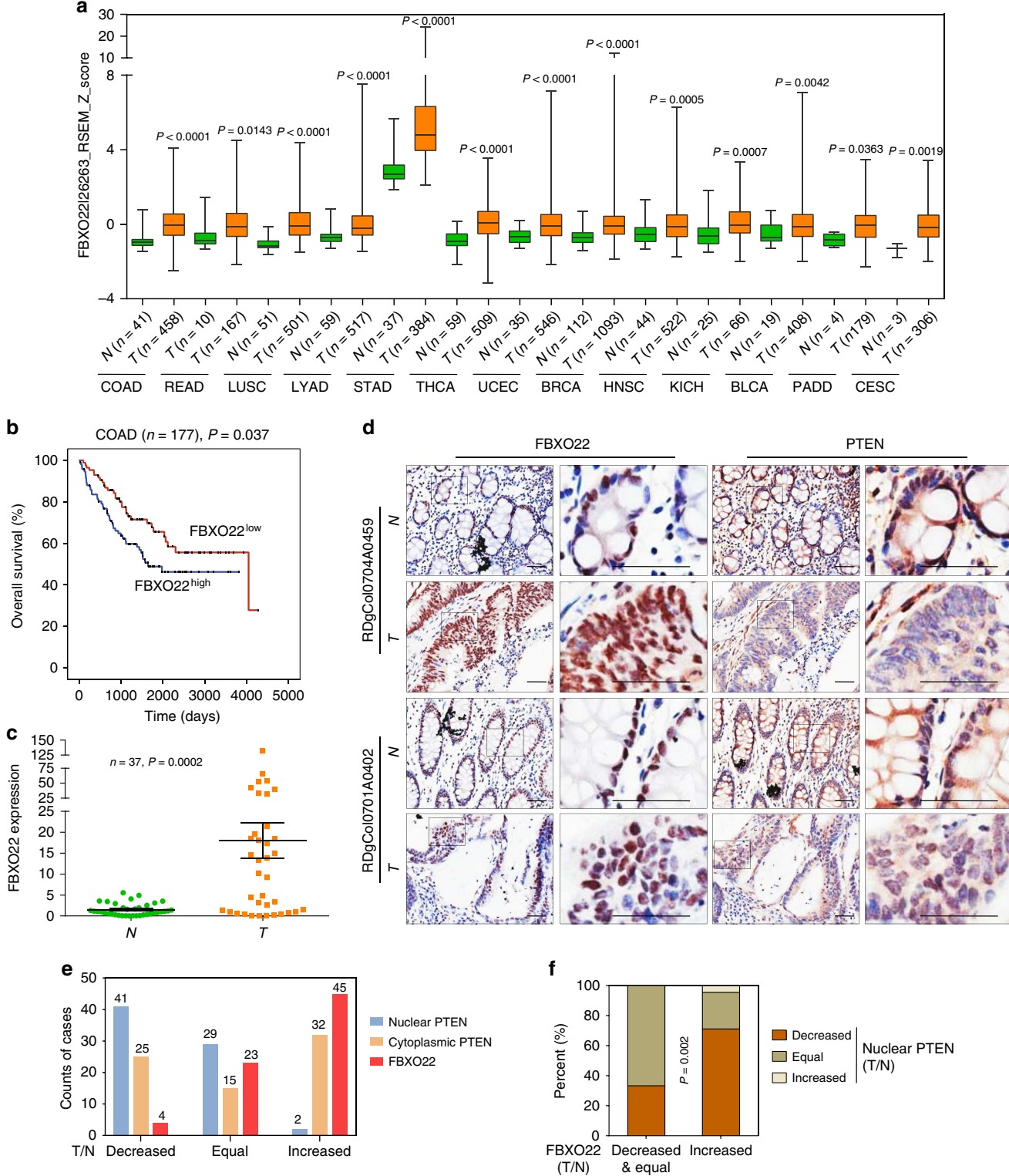

patients, and this absence of nuclear PTEN was correlated with the overexpression of FBXO22 in cancer tissues.

Besides canonical PTEN protein, recent studies also identified two evolutionarily conserved translational variants of PTEN, designated PTENα (PTEN-long or PTEN-L) and PTENβ[47–49]. They are respectively translated from a CUG site and an AUU site in the 5′-untranslated region of PTEN mRNA, thus containing an N-terminal extension of 173 and 146 amino acids followed by the 403 amino acids of PTEN. Some recent work showed that these

translational variants of PTEN present distinct functions from canonical PTEN protein. Whether FBXO22 also binds to and regulates PTENα/β in the nucleus warrants to be investigated in the future.

To date, cancer therapy and drug discovery efforts have predominantly focused on targeting oncogenic events, whereas the activation of tumor suppressors has remained less explored as a mode of cancer treatment[50]. As a potent tumor suppressor gene, PTEN was considered to be a good candidate for the long-sought

**Fig. 7 FBXO22 is overexpressed in cancers and negatively correlates with nuclear PTEN in colon cancer samples. a** Box and whisker plot showing z-scores of FBXO22 mRNA in TCGA cancer types with increased FBXO22 expression in tumor tissues compared with normal tissues. N normal; T tumor. **b** Kaplan–Meier estimates of overall survival of colon cancer patients with high and low FBXO22 expression in GEO GSE17536 dataset. **c** qRT-PCR analysis of FBXO22 expression in tumor tissues (T) and paired adjacent normal tissues (N) from a cohort of colorectal cancer patients. **d** Representative images of IHC analysis of FBXO22 and PTEN in the serial sections of tumor tissues (T) or paired adjacent normal tissues (N) isolated from two colorectal cancer patients ($n = 1$ experiment), with one representing patients with increased FBXO22 and the other representing patients with equal FBXO22 in T compared to N. Box areas in the left images were magnified on the right. Scale bar represents 50 μm. **e** Numbers of colorectal cancer patients respectively with decreased, equal or increased nuclear PTEN, cytoplasmic PTEN and FBXO22 in tumor tissues compared with paired adjacent normal tissues. The number of patients in each group is shown on top of the column. **f** Patient samples were divided into two groups respectively with decreased & equal or increased FBXO22 in tumor tissues compared with paired adjacent normal tissues, and the expressional pattern of nuclear PTEN in the two groups was analyzed. For **a**, the center line of the box and whisker plot represents the median, the boxes indicate the 25th and 75th centiles, and the whiskers indicate minimum and maximum values; statistical significance was determined by two-sided Mann–Whitney $U$ test. The data in **b** are analyzed by two-sided Mann–Whitney $U$ test. The data in **c** are presented as the the mean ± SEM, $n = 37$ biologically independent samples; statistical significance was determined by two-tailed unpaired $t$-test. The data in **f** are analyzed by Pearson's $\chi^2$ test. Source data are provided as a Source Data file.

"tumor suppressor reactivation" approach to cancer treatment. The recent discovery that MYC-WWP1 axis as a fundamental and evolutionary conserved regulatory pathway for PTEN and phosphatidylinositol-3-kinase (PI3K)-AKT signaling paved the way toward this end[27]. However, as usual, it only focused on the lipid phosphatase-dependent role of PTEN. Thus, our study reveals that nuclear and cytoplasmic PTEN not only exert distinct functions, but are also subjected to distinct regulatory mechanisms, and reports the first mechanism that preferentially regulates the stability of nuclear but not cytoplasmic PTEN. We believe that these findings might provide the opportunity for developing therapeutic strategies aiming to achieve complete reactivation of PTEN as a tumor suppressor.

## Methods

**Cell lines and cell culture.** Human cell lines including 293T, Hela, SW620, SW480, and LS174T were purchased from Cell Bank of Chinese Academy of Sciences, Shanghai. Mouse embryonic fibroblasts (MEF) were isolated from mouse embryo. 293T, Hela and MEF cells were maintained in Dulbecco's Modified Eagle's Medium (DMEM) supplemented with 10% fetal bovine serum (FBS) and the other three colorectal cancer cell lines in RPMI-1640 supplemented with 10% FBS. No signs of mycoplasma contamination were found for all cell lines. Short tandem repeat profiling was used for cell line authentication.

**Plasmids, shRNA, and antibodies.** Information on all of the plasmids is provided in Supplementary Table 1. Information on the shRNA sequences is provided in Supplementary Table 2. Antibodies employed are shown in Supplementary Table 3.

**Immunofluorescence.** Cells were seeded on coverslips, fixed in 4% paraformaldehyde at room temperature for 15 min, rinsed in PBS for three times, permeabilized in 0.3% Triton X-100 for 15 min, rinsed in PBS for three times, and blocked in 2% bovine serum albumin at room temperature for 1 h. Then, coverslips were incubated with primary antibodies 1:100 diluted in staining buffer (1% bovine serum albumin in 0.3% Triton X-100/PBS) overnight at 4 °C in a humid chamber. After washing three times, secondary antibodies (Alexa Fluor secondary 488, 595; Invitrogen) were applied in a 1:200 dilution in staining buffer for 1 h at 37 °C in a humid chamber in the dark. After washing, coverslips were mounted with Vectorshield with 4′,6-diamidino-2-phenylindole (DAPI; Vector Laboratories, CA), and analyzed on a Nikon Eclipse TI Laser Scanning Microscope or Leica TCS SP8.

**Subcellular fractionation.** NE-PER™ Nuclear and Cytoplasmic Extraction Reagents (Thermo Fisher Scientific) was used, and nuclear and cytoplasmic fractions were obtained following manufacturer's instructions.

**Western blot (immunoblotting).** Protein extracts were separated by SDS-PAGE, transferred to nitrocellulose membrane (Bio-Rad, Richmond, CA), blocked by 5% nonfat milk in PBS and sequentially incubated with the indicated primary antibodies and horseradish peroxidase (HRP)-linked secondary antibody (Cell Signaling, Beverly, MA), Immobilon Western Chemiluminescent HRP substrate kit (Merck Millipore) was used for detection. Microsoft PowerPoint 2016 and Photoshop CS6 were used to crop images from unprocessed images.

**In vivo ubiquitination assay.** Ubiquitylation assay was done following a denaturing IP protocol. The cells were treated with 10 μM of the proteasome inhibitor

MG132 (Calbiochem) for 6 h, washed with PBS, pelleted, submitted to subcellular fractionation and lysed with denature lysis buffer (denatured IP buffer 50 mM Tris-Cl, pH 6.8, 2% SDS). To dissociate protein-protein interactions, cell lysates were boiled for 15 min. After centrifugation at $12,000 \times g$ for 10 min at 16 °C, the supernatants were collected. Then, 70 μl of supernatantmixed with 1.2 ml RIPA buffer (50 mM Tris-HCl, PH 7.6, 150 mM NaCl, 1 mM EDTA, 1% NP-40, 1 mM PMSF, and 1 × protease inhibitor cocktail (Calbiochem) were subjected to immunoprecipitation with anti-FLAG M2 beads (Sigma-Aldrich) or PTEN antibody, followed by Western blot analysis to visualize polyubiquitylated protein bands.

**Nano-LC−ESI-MS/MS analysis.** Immunoprecipitation samples were separated by SDS–polyacrylamide gel electrophoresis, and visualized with colloidal Coomassie blue. The lane from gels was cut into 1 mm slices, and each slice was washed twice with 50 mM $NH_4HCO_3$, 50% acetontrile (ACN) and dehydrated with ACN. 10 mM DTT and 55 mM iodoacetamide were respectively used for protein reduction and alkylation. Proteins were then washed with 50 mM $NH_4HCO_3$ and ACN, subjected to in gel digestion with trypsin (Promega) overnight at 37 °C, followed by extraction of tryptic peptides from the gel pieces with 0.1% trifluoroacetic acid and 60% ACN. The extracted peptides were dried by vacuum centrifugation, and dissolved to 10 μl of 2% ACN and 0.1% trifluoroacetic acid. Afterwards, a Michrom peptide CapTrap (MW 0.5–50 kDa, 0.5 × 2 mm; Michrom BioResources, Inc.) was used to desalt and preconcentrate samples. The eluent was dried by vacuum and reconstituted in 2% ACN and 0.1% formic acid.

For analysis by LC–MS/MS, 10 μl samples were injected into a reverse-phase microcapillary column (0.1 × 150 mm, packed with 5 μm 100 Å Magic C18 resin; Michrom Bioresources) from an autosampler (HTS-PAL, CTC Analytics) at a flow of 1 μl/min for 15 min using a HPLC (Paragram MS4, Michrom Bioresources). The peptides were separated at a 0.5 ul/min flow rate usingbuffer A (2% ACN with 0.1% formic acid) and buffer B (98% ACN with 0.1% formic acid) with a 90 min gradient (0–35% B 90 min, 80% B 8 min, 95% B 12 min and 0% B 20 min). The MS spectrum was acquired using a hybrid linear ion trap (LTQ) Orbitrap mass spectrometer (ThermoFinnigan) assembled with ADVANCE Spray Source (Michrom Bioresources), using the following parameters. A high-resolution MS survey scan ($R = 100,000$ at m/z 400) was obtained for the m/z 350–1800 with $10^6$ ion accumulation of automatic gain control. Siloxane (m/z 445.120025) was used as an internal standard to calibrate the mass accuracy. Data-dependent scan determined by the Xcalibur mass spectrometer software was used to acquire MS/MS spectra from the 10 most intense ions within charge states 2–7. MS signals were at least 500 ion counts to trigger a MS/MS attempt and to acquire a MS/MS spectrum. Dynamic mass exclusion windows of 27 s were used. The normalized collision energy was set to 35%. ProteoWizard msConvert and Mascot (Matrix Science, London, UK; version 2.4.1) were respectively used to extract and analyze all MS/MS ion spectra. Mascot was set up to search the uniprot_human database (88,625 entries) based on the assumption that the digestion enzyme semiTrypsin allowed for two missed tryptic cleavages with full mass from 600 to 4600 with fragment ion mass tolerance being 1.00 Da and parent ion tolerance being 10.0 PPM. In Mascot, cysteine carbamidomethyl was specified as a fixed modification, and methionine oxidation and N terminus acetyl were specified as variable modifications. MS/MS-based peptide and protein identifications were validated by Scaffold (version Scaffold_4.2.1, Proteome Software Inc., Portland, OR) using the Scaffold Local FDR algorithm. Protein identifications were accepted at greater than 95.0% probability, which was assigned by the Protein Prophet algorithm. Proteins were grouped to satisfy the principles of parsimony if they contained similar peptides and could not be differentiated by MS/MS analysis alone. Proteins sharing significant peptide evidence were grouped into clusters. Proteins with quantitative values (normalized total spectra) in experimental group more than three times higher than control group were considered as potential interacting proteins. Other proteins identified were considered as non-specific binding.

**Immunoprecipitation**. Cells were harvested, lysed, and briefly sonicated. After centrifugation at 12,000 × g for 10 min at 4 °C, the supernatants (whole-cell lysates) were collected. For immunoprecipitation of Flag-tagged proteins or GFP-tagged proteins, cells were harvested and lysed after transfection and whole-cell lysates were incubated with anti-Flag M2 Affinity Gel or GFPTrap (ChromoTek) at 4 °C overnight. For immunoprecipitation of endogenous proteins, supernatants were sequentially incubated with PTEN antibodies overnight and protein A/G-agarose beads (Santa Cruz, CA) at 4 °C for 4 h. The precipitates were washed three times with immunoprecipitation buffer (50 mM Tris-HCl, PH 7.6, 150 mM NaCl, 1 mM EDTA, 1% NP-40, 1 mM PMSF, and 1x protease inhibitor cocktail (Calbiochem)), boiled in sample buffer and subjected to Western blot analysis.

**CRISPR-CAS9**. gFBXO22 sequences in the vector pLenti-U6-gRNA-mCMV-SaCas9-P-2A-sfGFP and gPTEN sequence in the vector pLenti-U6-spgRNA v2.0-CMV-Puro-P2A-3Flag-spCas9 were purchased from Obio Technology (Shanghai, China). The target sequences are provided in Supplementary Table 2. Cells were infected with virus carryingCAS9 and gRNA for 48 h. To obtain PTEN or FBXO22 knockout clones, cells were electroporated with the plasmid containing CAS9 and gRNA, subjected to selection with puromycin, followed by plating. The effect of gRNA was detected by immunoblotting. PCR and sequencing were used to confirm homozygous editing of the gene loci.

**Generation of Knock-in Flag-tagged FBXO22 cell line**. To generate the knock-in Flag-tagged FBXO22, a guide RNA targeting the first coding exon of FBXO22 (5′-TGGTGAGGAATGGAGCCGGT-3′) was cloned into px330-mcherry. To insert a Flag tag into this site via homologous recombination, we generated a single-stranded DNA as repair template with the Flag coding sequence surrounding the FBXO22 start codon. Cells were co-transfected with the gRNA bearing vectors and repair template. After 72 h, cells were sorted positive cells by FACS for mcherry fluorescence and plated into 96-well plates using a limited diluent method. Clones were screened for the insertion of the tag by PCR and validated by western blotting.

**qRT-PCR**. Total RNA was extracted using Trizol (Invitrogen) according to the manufacturer's instructions. RNA was digested with DNase I (Promega), reverse transcribed to cDNA using random primers (Takara) and M-MLV Reverse Transcriptase (Promega, Fitchburg, WI), followed by qRT-PCR with the SYBR Green PCR Master Mix (Applied Biosystems, Foster City, CA). Primers used in quantitative real-time PCR are presented in Supplementary Table 4.

**Cell cycle**. SW620 and SW480 cells were harvested, washed, and fixed with 75% cold ethanol at −20 °C overnight. Then, cells were treated with 100 µg/ml RNase A, stained with 25 µg/ml propidium ioide, and subjected to the analysis of cellular DNA content by flow cytometry (BD Biosciences) using FlowJo version 10 software. Ten thousand cells were acquired and analyzed for the DNA content.

**Luciferase/β-gal double-reporter assay**. The SW620 cells were electroporated with the double-reporter minigene pTN24 splicing reporter plasmid[51], treated with doxycycline or DMSO 72 h and harvested. The Dual Light Reporter System (Applied Biosystems) was used to detect reporter expression, followed by calculation of the the luciferase to β-galactosidase ratio.

**Mouse studies**. SW620 and LS174T, respectively, were inoculated subcutaneously into female nude mice and NOD-SCID mice (Shanghai Laboratory of Animal Center, Chinese Academy of Sciences) and tumor volume was monitored. Mice were administrated with or without 600 mg/kg DOX in feed immediately after subcutaneous injection[46]. Mice were housed in groups (3–5 per cage) at 22–24 °C with a 12 h light-dark cycle and ad libitum access to regular chow diet and water. Animal care and experiments were in agreement with all of the animal research related ethical regulationsunder the approvement ofthe committee for humane treatment of animals at Shanghai Jiao Tong University School of Medicine.

**Dataset analysis**. The RNA expression data and related survival data were retrieved from public datasets (The Cancer Genome Atlas Research Network, 2015, and Gene Expression Omnibus (GEO)). For analysis of overall survival in TCGA datasets, 35% tumor tissues with highest and lowest FBXO22 expression were compared. For analysis in GEO GSE17536 dataset, tumor tissues were divided in halves according to FBXO22 expression.

**Human colorectal cancer samples**. Human colorectal cancer tissues and paired normal tissues for qRT-PCR and Immunoblotting analysis were randomly selected with informed consent from human tissue bank of Ruijin Hospital of Shanghai Jiao Tong University School of Medicine (SJTU-SM) under approval of the Medical Ethic Committee of SJTU-SM.

Tissue microarray of human colorectal tumors and paired adjacent normal tissues were purchased from Shanghai Outdo Biotech Co (Shanghai, China). The immunoreactive score (IRS) gives a range of 0–12 as a product of multiplication between positive cells proportion score (0–4) and staining intensity score (0–3).

The diagnosis of normal tissue or colorectal cancer was confirmed by independent pathologists based on histological findings. All experiments were performed with informed consent obtained from all subjects with the approval of the Medical Ethic Committee of Shanghai Jiao Tong University School of Medicine Review Board.

**Statistical analysis**. As described in figure legends, the statistical analyses were performed using the two-tailed unpaired Student's $t$-test, Pearson's $\chi^2$ test, Two-way ANOVA, log-rank test and two-sided Mann–Whitney $U$ test using GraphPad Prism 7 (GraphPad Software), Microsoft Excel 2016, and IBM SPSS Statistics 20. $P < 0.05$ was considered statistically significant. As indicated in the figure legends, experiments were repeated independently multiple times and similar results were obtained.

**Reporting summary**. Further information on research design is available in the Nature Research Reporting Summary linked to this article.

## Data availability

The data supporting the findings of this manuscript are available from the corresponding author upon reasonable request. Previously published RNA-Seq data that were reanalyzed here are available under accession code GSE17536. The mass spectrometry proteomics data have been deposited to the ProteomeXchange Consortium via the PRIDE partner repository with the dataset identifier PXD017983 and 10.6019/PXD017983. The source data underlying Figs. 1b–d, 2a–i, 3a–c, e–g, 4a–e, g–j, 5a–e, 6a–f, 7c, f and Supplementary Figs. 1a, d, 2a, b, d, f, h, 3a–d, 4a–g, 5b–d are provided as a Source Data file.

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

## Acknowledgements

This work was supported by National Natural Science Foundation (81830091, 91853206, 81972583) and its innovative group support (No.81721004), CAMS Innovation Fund for Medical Sciences (CIFMS) (2019-I2M-5-051) as well as the Fundamental Research Funds for the Central Universities.

## Author contributions

M.-K.G., N.Z., and L.X. performed most experiments. C.Z., S.-S.D., Z.-M.L., and Y.J. conducted partial experiments. M-H.Z. and J.S. collected patient samples. G-Q.C. and S.-M.S. designed and supervised the entire project and prepared the paper.

## Competing interests
The authors declare no competing interests.
