## [Peer Review File · Nature Communications]

Reviewers' comments:

Reviewer #1 (Remarks to the Author):

In this manuscript, Ge et al. demonstrate that FBXO22 facilitates ubiquitination of nuclear PTEN and degrades nuclear PTEN. The authors also provide evidence that FBXO22 blocks nuclear PTEN signaling and promotes tumorigenesis via degradation of nuclear PTEN. The paper provides new insight into the regulatory modes of nuclear PTEN and the role of nuclear PTEN in colon cancer. However, the pathologic evidence is not solid enough to demonstrate the role of FBXO22 in tumorigenesis, and several aspects of the manuscript need improvement.

1. The tumor suppressive function of nuclear PTEN was not clear in colon cancer, until it was reported that nuclear PTEN stabilizes RPA1 and protects the genome. It should be explained why the authors chose colon cancer as the initial model for study of nuclear PTEN. When I checked the TCGA database to confirm the role of FBXO22 in different types of cancer, I found the results are much different from what are shown in Fig. 7B and Fig. S5 A-C. According to the TCGA database, expression levels of FBXO22 are not prognostic in colon cancer (<https://www.proteinatlas.org/ENSG00000167196-FBXO22/pathology/colorectal+cancer/COAD>), glioma (<https://www.proteinatlas.org/ENSG00000167196-FBXO22/pathology/glioma>) and breast cancer (<https://www.proteinatlas.org/ENSG00000167196-FBXO22/pathology/breast+cancer>). Thus, the authors should describe Fig. 7B and Fig. S5 A-C in much more detail.
2. In figure 2A, the association between CUL1, SKP1 and PTEN should be validated.
3. The reference list of this manuscript should be updated. The references are not comprehensive enough as the list is short. For example, when mentioning the post-transcriptional and post-translational modifications of PTEN, recent important progresses in the PTEN field should be cited.
4. FBXO22 should be introduced in details in the INTRODUCTION section.

Reviewer #2 (Remarks to the Author):

The authors seek to understand the molecular mechanism underlying how FBXO22-mediated degradation of nuclear PTEN promotes tumorigenesis. This manuscript utilized a combined biochemical and genetic approach to identify PTEN as a novel ubiquitin substrate of FBXO22, which underlies the oncogenic role of FBXO22. The paper is clearly written, however, the following minor concerns should be addressed before its publication at Nature Communications.

1. Some typos, such as in page 5, duplicated "," and one should be deleted in line three of Results section.
2. Figure 1B, the authors should also consider the possibility that MG132 might promote the nuclear translocation of PTEN as PTEN mono-ub has been implicated in its nuclear translocation process.
3. Figure 1C, the authors should define which ub linkage was largely added on nuclear vs cytoplasmic PTEN? Are they different, which causes the different stability phenotypes for cytoplasmic vs nuclear PTEN?
4. Figure S1B, have the authors tested WWP1?
5. Page 8, FBXO22 is not an orphan F-box protein as Bach1 has been identified as Fbxo22 substrate.
6. Figure 2D, will Fbxo22 be deficient to degrade nuclear PTEN but capable of ubiquitinating cytoplasmic PTEN if the authors delete the NLS of FBXO22 or adds a NES.
7. Figure 3E, it will be nice for the authors to use either NLS-deleted or NES added Fbxo22 as a control.
8. Figure 3G, WWP2 has been previously identified as an E3 ligase to degrade PTEN. It will be

important to side by side compare WWP2KO with FBXO22 KO cell lines. The authors found only accumulation of nuclear PTEN upon Fbxo22 KO, how about WWP2 KO, will it lead to accumulation of cytoplasmic PTEN?

9. Figure 4C, it is important to show that APC/Cdh1 is the important pathway for this phenotype. Will ectopic expression of Cdh1 rescue the phenotypes?

10. Figure S3A, the authors should explain why ectopic expression of NEDD4 elevates PIK1 levels?

11. Figure 5B-C, the authors should examine whether K221R mutation affects PTEN nuclear localization and also affect the half-life of nuclear PTEN.

12. Figure 6C and 6F, pAkt, cyclin A and PIK1 should be monitored to show whether decrease in expression of these proteins, and not pAkt changes underlie the retarded tumorigenesis in vivo.

Reviewer #3 (Remarks to the Author):

In this manuscript, the authors present data supporting the hypothesis that FbxO22 degrades nuclear PTEN to promote tumorigenesis. The main findings are that 1) nuclear and cytoplasmic PTEN are differentially regulated, 2) nuclear PTEN is degraded at the proteasome, 3) FbxO22 ubiquitinates PTEN promoting proteasomal degradation at lysine 221 of PTEN, and 4) FbxO22 promotes tumorigenesis. Data is presented in HEK cells, human colorectal cancer cell lines, mouse embryonic fibroblasts, a mouse model and samples from a human tumor repository. Overall, the studies demonstrate a unique molecular model of regulatory effect that is compartment specific for SCF ligases. The data are generally internally consistent with some good use of controls. The largely genetic approach is robust in multiple cell lines using knockout and overexpression. Finally, the human data correlates with the story. The main limitations of the study are that Fbxo22 has several known substrates and these are not assessed as controls, and second, the molecular signatures that partake in Fbxo22 recruitment to PTEN are not demonstrated.

Major:

1) Phosphopeptidic signatures, acetylation, and glycosylation marks on substrates recruit F-box proteins. This should be assessed or docking regions between Fbxo22 and PTEN demonstrated.

2) A rescue experiment where depletion of Fbxo22 and reversal of its abundance on the target should be tested.

3) Other known clients of Fbxo22 should be shown as positive controls.

4) Does the K acceptor (K221) site mutant bind Fbxo22?

Other:

Introduction- page 4 bottom-Fbxo22 is not an orphan protein as there are numerous clients

Fig 1B: While I appreciate the effect of proteasomal inhibitors on PTEN, the "lysosomal inhibitors" are interesting choices. NH₄Cl (alternates are bafilomycin and chloroquine) is an endosome and lysosome acidification inhibitor and a commonly used reagent for this purpose. Brefeldin A (BFA) inhibits trafficking between the Golgi and ER with indirect lysosome effects. A protease inhibitor, such as leupeptin, is a better control here. There may be a weak effect with NH₄Cl compared to DMSO control, but the MG-132 and bortezomib effects are certainly more impressive.

Fig S2B: Please show the separate channels (Green and DAPI separate + merge).

Fig S3C. Data is not convincing. I see not change in CyclinA2 between 0-90 hrs.

Fig 2: Overall, the biochemical work demonstrating PTEN ubiquitination is internally consistent. However, all of it is done with overexpression of PTEN, Ubiquitin and often FbxO22. Understanding that western blotting of ubiquitination is difficult, especially in some of the experiments presented requiring multiple manipulations, the authors should include an experiment showing endogenous

ubiquitination, perhaps with FbxO22 knockout or overexpression. It appears from Fig 2A that you get robust pulldown of PTEN. An alternate approach would be to pull down ubiquitinated proteins with an antibody or TUBEs to look for Ub-PTEN.

Fig 2: Is PTEN K48 or K63 ubiquitinated in this system? Interestingly NEDD4-1 and WWP2 are thought to preferentially K63 ubiquitinate proteins, whereas FbxO22 ubiquitinates proteins by K48-linked conjugation. This can be tested by PTEN pulldown followed by probing with antibodies or TUBEs reagents specific to these linkages (or vice versa).

Fig 5: Consider adding a cycloheximide dose chase with PTEN K221R to show differences in stability. It would also be interesting to explore the linkage specificity on K221.

Point-by-point responses to reviewers' comments

Responses for Reviewer #1:

Summary: In this manuscript, Ge et al. demonstrate that FBXO22 facilitates ubiquitination of nuclear PTEN and degrades nuclear PTEN. The authors also provide evidence that FBXO22 blocks nuclear PTEN signaling and promotes tumorigenesis via degradation of nuclear PTEN. The paper provides new insight into the regulatory modes of nuclear PTEN and the role of nuclear PTEN in colon cancer. However, the pathologic evidence is not solid enough to demonstrate the role of FBXO22 in tumorigenesis, and several aspects of the manuscript need improvement.

RESPONSE: We appreciate the reviewer's kind evaluations on our manuscript. In this version, we have enforced the pathologic evidence to demonstrate the role of FBXO22 in tumorigenesis, and several aspects proposed by this reviewer have been greatly improved.

Comment 1: The tumor suppressive function of nuclear PTEN was not clear in colon cancer, until it was reported that nuclear PTEN stabilizes RPA1 and protects the genome. It should be explained why the authors chose colon cancer as the initial model for study of nuclear PTEN. When I checked the TCGA database to confirm the role of FBXO22 in different types of cancer, I found the results are much different from what are shown in Fig. 7B and Fig. S5 A-C. According to the TCGA database, expression levels of FBXO22 are not prognostic in colon cancer (<https://www.proteinatlas.org/ENSG00000167196-FBXO22/pathology/colorectal+cancer/COAD>), glioma (<https://www.proteinatlas.org/ENSG00000167196-FBXO22/pathology/glioma>) and breast cancer (<https://www.proteinatlas.org/ENSG00000167196-FBXO22/pathology/breast+cancer>). Thus, the authors should describe Fig. 7B and Fig. S5 A-C in much more detail.

RESPONSE: Thanks. As we mentioned on the lines 212-216 of page 10 of the revised version, the tumor suppressive function of nuclear PTEN in colon cancer has been previously explored, and genetic alterations of PTEN gene is relatively rare in colon cancer (ref. 28), suggesting the regulation of PTEN more likely relies on mechanisms beyond genetic alterations. Thus, we chose

colon cancer as a model for our further study. Of course, here we also cited the reference (ref#19, Wang, G. et al. Cell research 2015; 25, 1189-1204). Notably, as shown in current Fig. 7a, colon cancer is also one of the cancer types with most significant up-regulation of FBXO22 in tumor tissues compared to normal tissues, which is also one of reasons why we selected colon cancer as our model.

By the way, this version provided some explanations in detail in METHODS section for some of the data processing steps in current Fig. 7b and Supplementary Fig. 6a-d. Shortly, the original data on FBXO22 and patient survival in TCGA and other datasets were downloaded. For TCGA datasets, 35% highest and 35% lowest expression of FBXO22 were used for each cancer type due to the large sample size to analyze correlation between FBXO22 and patient survival in these datasets. Indeed, our analysis showed that there are inverse correlations between FBXO22 and patient survival in glioma (Supplementary Fig. 6a), pancreatic adenocarcinoma (Supplementary Fig. 6b) and invasive breast cancer (Supplementary Fig. 6c) datasets. Indeed, such an inverse correlation could not be seen in TCGA colorectal cancer dataset (current Supplementary Fig. 6d). However, we did show an inverse correlation between FBXO22 and patient survival in GEO colon cancer dataset GSE17536 (Fig. 7b), in which high and low FBXO22 expression were divided in halves because of the relatively small sample size. The relative descriptions can also be found in their legends.

Comment 2: In figure 2A, the association between CUL1, SKP1 and PTEN should be validated.

RESPONSE: Thanks. The current Fig. 2a showed that anti-PTEN antibody could pull down CUL1 and SKP1 besides FBXO22 and PTEN. Also, we added current Supplementary Fig. 2a into the revised version, which showed that CUL1-Flag could precipitate PTEN, FBXO22 and SKP1. All these data support the association between CUL1, SKP1 and PTEN.

Comment 3: The reference list of this manuscript should be updated. The references are not comprehensive enough as the list is short. For example, when

mentioning the post-transcriptional and post-translational modifications of PTEN, recent important progresses in the PTEN field should be cited.

RESPONSE: Thanks, some recent important and related progresses in the PTEN field have been cited, which made references from the previous 35 to current 52.

Comment 4: FBXO22 should be introduced in details in the INTRODUCTION section.

RESPONSE: Ok. We have described it in the last paragraph of INTRODUCTION section of the revised version.

Responses for Reviewer #2:

Summary: The authors seek to understand the molecular mechanism underlying how FBXO22-mediated degradation of nuclear PTEN promotes tumorigenesis. This manuscript utilized a combined biochemical and genetic approach to identify PTEN as a novel ubiquitin substrate of FBXO22, which underlies the oncogenic role of FBXO22. The paper is clearly written, however, the following minor concerns should be addressed before its publication at Nature Communications.

RESPONSE: Thanks for the reviewer's good summary on this study. We have revised the manuscript according to your comments.

Comment 1: Some typos, such as in page 5, duplicated “,” and one should be deleted in line three of Results section.

RESPONSE: Thanks! We have done our best to correct such typos throughout.

Comment 2: Figure 1B, the authors should also consider the possibility that MG132 might promote the nuclear translocation of PTEN as PTEN mono-ub has been implicated in its nuclear translocation process.

RESPONSE: Considering this comment, we also transfected GFP-tagged K13 and K289 mutated PTEN (PTEN^{K13, 289E}) into 293T cells, because monoubiquitylation of PTEN on these two lysines was reported to regulate its nuclear import (ref#22). Our results showed that MG132 treatment still

increased PTEN^{K13, 289E}-GFP in the nucleus to similar degree as wild-type PTEN (current Supplementary Fig. 1b). Therefore, this result should exclude the possibility that MG132 might promote the nuclear translocation of PTEN as PTEN mono-Ub. The related description has been added in the revised version (lines 116-121 of page 6).

Comment 3: Figure 1C, the authors should define which ub linkage was largely added on nuclear vs cytoplasmic PTEN? Are they different, which causes the different stability phenotypes for cytoplasmic vs nuclear PTEN?

RESPONSE: To reply the concern, we used antibodies against K48 or K63 Ub chain to detect the Ub linkage of PTEN. Interestingly, we show that PTEN was predominantly ubiquitylated by K48 but not K63 Ub chain (current Fig. 1d), whereas MG132 treatment accumulated nuclear but not cytoplasmic K48 Ub chain linked PTEN. The related description has been added in the revised version (lines 133-137 of page 6). Therefore, as the pattern of PTEN ubiquitylation by K48 Ub chain is similar to that of total Ub, we propose that K48 Ub chain contributes to the different stability phenotypes for nuclear and cytoplasmic PTEN. This is consistent with the notion that K48-linked poly-Ub chains are most commonly associated with proteins targeted for proteasomal degradation.

Comment 4: Figure S1B, have the authors tested WWP1?

RESPONSE: Considering that WWP1 was reported to specifically trigger nondegradative K27-linked polyubiquitination of PTEN to suppress its dimerization, membrane recruitment, and tumor suppressive functions as stated in the manuscript (ref#27), previously we did not include it as a control, since FBXO22 triggers degradation of nuclear PTEN. Here we show that GFP-tagged WWP1 is mainly cytoplasmic, like GFP-tagged NEDD4-1 and WWP2 (current Fig. 1e). By the way, we could not detect localization of endogenous WWP1 due to unavailability of anti-WWP1 antibody for immunofluorescence.

Comment 5: Page 8, FBXO22 is not an orphan F-box protein as Bach1 has been identified as Fbxo22 substrate.

RESPONSE: Thanks. We have corrected it, and added the corresponding references for BACH1 into the revised text.

Comment 6: Figure 2D, will Fbxo22 be deficient to degrade nuclear PTEN but capable of ubiquitinating cytoplasmic PTEN if the authors delete the NLS of FBXO22 or adds a NES.

RESPONSE: Good suggestion! To answer this question, we found a highly probable sequence, K¹²¹RARKR¹²⁶, to be essential for the nuclear localization of FBXO22 (current Fig. 3d). Thus, we expressed NLS mutated FBXO22 (FBXO22^{mNLS}) alongside wild-type FBXO22 (FBXO22^{WT}) in SW620 cells, and found that while FBXO22^{WT} overexpression still down-regulated nuclear but not cytoplasmic PTEN, cytoplasm-localized FBXO22^{mNLS} overexpression affected neither nuclear nor cytoplasmic PTEN (current Fig. 3e). Further analysis revealed that FBXO22^{mNLS} did not interact with PTEN (current Fig. 3f), which resulted in the inability of FBXO22^{mNLS} to degrade cytoplasmic PTEN. The related description has been added in the text (lines 232-243 of page 10).

Comment 7: Figure 3E, it will be nice for the authors to use either NLS-deleted or NES added Fbxo22 as a control.

RESPONSE: Thanks. We think our experiments to reply your comment above (comment 6) can make the conclusion from previous Fig. 3E (current Supplementary Fig. 3c) nicer.

Comment 8: Figure 3G, WWP2 has been previously identified as an E3 ligase to degrade PTEN. It will be important to side by side compare WWP2KO with FBXO22 KO cell lines. The authors found only accumulation of nuclear PTEN upon Fbxo22 KO, how about WWP2 KO, will it lead to accumulation of cytoplasmic PTEN?

RESPONSE: In the current Supplementary Fig. 3a, we further showed that unlike FBXO22, knockdown of NEDD4-1 and WWP2 by their specific shRNAs only increased cytoplasmic PTEN without influencing nuclear PTEN.

Comment 9: Figure 4C, it is important to show that APC/Cdh1 is the important pathway for this phenotype. Will ectopic expression of Cdh1 rescue the phenotypes?

RESPONSE: Good suggestion! For this purpose, we have constructed a Cdh1-expressing plasmid, and stably expressed Cdh1 or empty control (EV) in SW620 cells with inducible expression of FBXO22. We show that the levels of Aurora A, PLK1 and Cyclin A2 increased by FBXO22 induction in EV-expressing cells, but failed to increase in Cdh1-expressing cells (current Supplementary Fig. 4d). Thus, APC/Cdh1 is indeed important for the action of FBXO22 here (lines 270-273 of page 12).

Comment 10: Figure S3A, the authors should explain why ectopic expression of NEDD4 elevates PIK1 levels?

RESPONSE: Thanks. In this version, we mentioned that “It is worthwhile to point out that knockdown of NEDD4-1 increased the level of PLK1 (Supplementary Fig. 4c), but its mechanisms remain to be illustrated” on lines 266-269 of page 12.

Comment 11: Figure 5B-C, the authors should examine whether K221R mutation affects PTEN nuclear localization and also affect the half-life of nuclear PTEN.

RESPONSE: Thanks. We have performed these assays and found that K221R mutation does not significantly affect PTEN nuclear localization (current Supplementary Fig. 5e), and PTEN^{K221R} is more stable than PTEN^{WT} in the nucleus upon CHX treatment (current Fig. 5c). The related description has been added in the text (lines 321-325 of page 14).

Comment 12: Figure 6C and 6F, pAkt, cyclin A and PIK1 should be monitored to show whether decrease in expression of these proteins, and not pAkt changes underlie the retarded tumorigenesis in vivo.

RESPONSE: We have done it in the revised version (current Fig. 6c, f). The results showed that Aurora A, PLK1 and Cyclin A2 were not as efficiently induced by FBXO22 overexpression in PTEN^{K221R} tumors as in PTEN^{WT} tumors, whereas

pAKT was comparable between them, suggesting it was the changes of Aurora A, PLK1 and Cyclin A2 rather than pAKT that underlie the retarded tumorigenesis in vivo. The related description has been added in the text (lines 351-359 of page 15).

Responses for Reviewer #3:

Summary: In this manuscript, the authors present data supporting the hypothesis that FbxO22 degrades nuclear PTEN to promote tumorigenesis. The main findings are that 1) nuclear and cytoplasmic PTEN are differentially regulated, 2) nuclear PTEN is degraded at the proteasome, 3) FbxO22 ubiquitinates PTEN promoting proteasomal degradation at lysine 221 of PTEN, and 4) FbxO22 promotes tumorigenesis. Data is presented in HEK cells, human colorectal cancer cell lines, mouse embryonic fibroblasts, a mouse model and samples from a human tumor repository. Overall, the studies demonstrate a unique molecular model of regulatory effect that is compartment specific for SCF ligases. The data are generally internally consistent with some good use of controls. The largely genetic approach is robust in multiple cell lines using knockout and overexpression. Finally, the human data correlates with the story. The main limitations of the study are that Fbxo22 has several known substrates and these are not assessed as controls, and second, the molecular signatures that partake in Fbxo22 recruitment to PTEN are not demonstrated.

RESPONSE: We appreciate your good evaluations and constructive comments on this study. We have revised the manuscript according to your comments, which should greatly strengthen the work.

Major 1: Phosphopeptidic signatures, acetylation, and glycosylation marks on substrates recruit F-box proteins. This should be assessed or docking regions between Fbxo22 and PTEN demonstrated.

RESPONSE: According to the concern, we assessed docking regions of PTEN for its interaction with FBXO22. Considering the differences in subcellular localization of PTEN fragments might interfere with the outcome, we sought to put different PTEN derivatives in the same cellular compartment to compare their interaction with FBXO22. Phosphorylation of PTEN on S380/T382/T383/S385 has been reported to regulate PTEN nuclear

localization, as their concomitant mutation (PTEN^{4A}) almost completely relocalized PTEN into the nucleus (ref#43). Indeed, we verified this phenomenon in our system (current Supplementary Fig. 2e). We also show that endogenous PTEN phosphorylated on S380/T382/T383 are almost exclusively cytoplasmic (current Supplementary Fig. 2d). In accordance, the nuclear localized PTEN mutant PTEN^{4A} showed much more interaction with FBXO22 than wild-type PTEN (PTEN^{WT}) (current Supplementary Fig. 2f). We also generated a C-tail deleted fragment of PTEN, PTEN^{1~351} (current Supplementary Fig. 2g), which also showed similar dominant nuclear distribution as PTEN^{4A} (current Supplementary Fig. 2e). Although both were nuclear localized, PTEN^{1~351} showed much weaker interaction with FBXO22 than PTEN^{4A} (current Supplementary Fig. 2h), indicating a role for PTEN C-tail. Based on this findings, amino acids were gradually added to the C-terminus of PTEN^{1~351}, and we found that amino acids 366-375 was critical for PTEN-FBXO22 interaction (current Fig. 2i and Supplementary Fig. 2h). For detail, please refer to lines 187-203 of page 9. By the way, there are two known phosphorylated sites (T366, S370) in 366-375 region of PTEN. Whether their phosphorylations are required for FBXO22-PTEN interaction remains to be investigated in the future.

Major 2: A rescue experiment where depletion of Fbxo22 and reversal of its abundance on the target should be tested.

RESPONSE: We have done it in current Supplementary Fig. 4a, which showed that re-expression of an shRNA-resistant version of FBXO22 could reverse FBXO22 knockdown-induced the increase of PTEN and the decreases of PLK1, Aurora A and Cyclin A2. The related description have been added in the text (lines 262-263 of page 12).

Major 3: Other known clients of Fbxo22 should be shown as positive controls.

RESPONSE: Thanks. In this version we used the BACH1 as a positive control for FBXO22 substrate, and showed that FBXO22 knockdown and inducible expression could elevate and decrease BACH1 and nuclear PTEN (see current Fig. 3a-c,4a,4b and Supplementary Fig. 3a).

Major 4: Does the K acceptor (K221) site mutant bind Fbxo22?

RESPONSE: We showed that K221 mutant almost completely abolished K48 ubiquitin chain linked PTEN ubiquitylation by FBXO22 (Fig. 5a, b), without affecting the interaction between PTEN and FBXO22 (Supplementary Fig. 5d).

Others 1: Introduction- page 4 bottom-Fbxo22 is not an orphan protein as there are numerous clients.

RESPONSE: Thanks! We have corrected it.

Others 2: Fig 1B: While I appreciate the effect of proteasomal inhibitors on PTEN, the “lysosomal inhibitors” are interesting choices. NH₄Cl (alternates are bafilomycin and chloroquine) is an endosome and lysosome acidification inhibitor and a commonly used reagent for this purpose. Brefeldin A (BFA) inhibits trafficking between the Golgi and ER with indirect lysosome effects. A protease inhibitor, such as leupeptin, is a better control here. There may be a weak effect with NH₄Cl compared to DMSO control, but the MG-132 and bortezomib effects are certainly more impressive.

RESPONSE: Thanks, we have modified the related descriptions in paragraph 1 of results section “Nuclear PTEN is sensitive to ubiquitin-mediated proteasomal degradation”, and also pointed out that MG132 and Bortezomib abundantly accumulated nuclear PTEN without affecting cytoplasmic PTEN in both cell lines, although the lysosome inhibitor NH₄Cl or the endoplasmic reticulum to Golgi transport inhibitor Brefeldin A showed a weak or no effect on nuclear PTEN accumulation.

Others 3: Fig S2B: Please show the separate channels (Green and DAPI separate + merge).

RESPONSE: We have done it (current Supplementary Fig. 2c).

Others 4: Fig S3C. Data is not convincing. I see not change in CyclinA2 between

0-90 hrs.

RESPONSE: Indeed, in LS174T cell line, when FBXO22 was overexpressed, Cyclin A2 only showed a mild increase (current Supplementary Fig. 4e). This phenomenon was repeatedly observed in this specific cell line. We reasoned that the basal level of Cyclin A2 in LS174T cell line is too high to be further increased. To be more accurate, we have modified the related description in the revised version (lines 274-276 of page 12).

Others 5: Fig 2: Overall, the biochemical work demonstrating PTEN ubiquitination is internally consistent. However, all of it is done with overexpression of PTEN, Ubiquitin and often FbxO22. Understanding that western blotting of ubiquitination is difficult, especially in some of the experiments presented requiring multiple manipulations, the authors should include an experiment showing endogenous ubiquitination, perhaps with FbxO22 knockout or overexpression. It appears from Fig 2A that you get robust pulldown of PTEN. An alternate approach would be to pull down ubiquitinated proteins with an antibody or TUBEs to look for Ub-PTEN.

RESPONSE: Good suggestion! We have performed ubiquitination assay of endogenous PTEN with endogenous Ubiquitin, by FBXO22 overexpression (current Fig. 2e). The related description has been added in the text (lines 168-169 of page 8).

Others 6: Fig 2: Is PTEN K48 or K63 ubiquitinated in this system? Interestingly NEDD4-1 and WWP2 are thought to preferentially K63 ubiquitinate proteins, whereas FbxO22 ubiquitinates proteins by K48-linked conjugation. This can be tested by PTEN pulldown followed by probing with antibodies or TUBEs reagents specific to these linkages (or vice versa).

RESPONSE: As the reviewer suggested, we detected FBXO22-induced PTEN ubiquitination with antibodies against K48 or K63 ubiquitin chain, and our results showed that FBXO22 ubiquitinated PTEN by K48-linked conjugation (current Fig.2d and Fig.5b). It is consistent with the notion K48-linked polyubiquitin chains are most commonly associated with proteins targeted for proteasomal degradation and K63-linked polyubiquitin chains are associated with nonproteolytic functions.

Others 7: Fig 5: Consider adding a cycloheximide dose chase with PTEN K221R to show differences in stability. It would also be interesting to explore the linkage specificity on K221.

RESPONSE: Good suggestion! To do so, SW620 cells stably expressing PTEN wild-type (PTEN^{WT}) or K221R (PTEN^{K221R}) were subjected to a CHX dose chase, followed by cellular fractionation. The result showed that while both cytoplasmic PTEN^{WT} and PTEN^{K221R} remained stable during 12 h's CHX treatment, the nuclear portion of PTEN^{WT} was more unstable than that of PTEN^{K221R} (current Fig. 5c). The related description has been added in the text (lines 323-325 of page 14).

To explore the linkage specificity on K221, we detected FBXO22-induced PTEN^{WT} and PTEN^{K221R} ubiquitination with antibodies against K48 or K63 ubiquitin chain. We found that FBXO22 ubiquitylated PTEN^{WT} mainly with K48 but not K63 ubiquitin chain, and K221 mutation significantly abolished this modification (current Fig. 5b). It is consistent with the notion K48-linked polyubiquitin chains are most commonly associated with proteins targeted for proteasomal degradation and K63-linked polyubiquitin chains are associated with nonproteolytic functions, since K221 ubiquitination of PTEN by FBXO22 leads to proteasomal degradation. Thus, our results showed that FBXO22 ubiquitinates K221 with K48 ubiquitin chain.

REVIEWERS' COMMENTS:

Reviewer #1 (Remarks to the Author):

The authors have adequately addressed the issues that I raised.

Please note that PTEN recruits OTUB1, but not OTUD1, to deubiquitinates RPA1 (line 59).

I have also carefully read the comments made by Reviewer #3 and the related response made by the authors, as the editor invited me to do so. I think the authors have addressed all issues that the Reviewer #3 raised.

Reviewer #2 (Remarks to the Author):

The authors have addressed most of the raised concerns during this round of revision.

Point-by-point responses to reviewers' comments

Responses for Reviewer #1

Comment: Please note that PTEN recruits OTUB1, but not OTUD1, to deubiquitinates RPA1 (line 59).

Response: Thanks. We have changed OTUD1 to OTUB1 in the revised version (line 59 of page 3).